



# Spatio-temporal snow data assimilation with the ICESat-2 laser altimeter

Marco Mazzolini[1], Kristoffer Aalstad[1], Esteban Alonso-González[2], Sebastian Westermann[1], and Désirée Treichler[1]

[1]Department of Geosciences, University of Oslo, Oslo, Norway
[2]Pyrenean Institute of Ecology, CSIC, Jaca, Spain.
**Correspondence:** Marco Mazzolini (marcomaz@uio.no)

**Abstract.** The satellite laser altimeter ICESat-2 provides accurate surface elevation observations across the globe. Where a high-resolution DEM is available, we can use these measurements to retrieve snow depth profiles even in areas where snow amounts are poorly constrained, despite being of great societal interest. However, the adoption of these retrievals remains low since they are very sparse in space (the satellite measures along profiles) and in time (the revisit is 3 months). Data assimilation

methods can exploit snow observations to constrain snow models and provide gap-free snow map time series. Assimilation of observations like snow cover is established, but there are currently no methods to assimilate sparse ICESat-2 snow depth profiles. We propose an approach that spatially propagates information using – instead of the classic geographical distance – an abstract distance measured in a feature space defined by topographical parameters and the melt-out climatology.

We demonstrate this framework for a small experimental catchment in the Spanish Pyrenees through three experiments. We

assimilate different observations in an intermediate-complexity snow model: fractional snow cover retrievals from Sentinel-2, snow depth profiles from ICESat-2 located in proximity of the catchment, or both snow cover and depth in a joint assimilation experiment. Results show that assimilating ICESat-2 snow depth profiles successfully updates the neighboring *unobserved* catchment, improving the simulated average snow depth compared to the prior run. Moreover, adding the snow depth profiles to fractional snow-covered area observations leads to an accurate reconstruction of the snow depth spatial distribution, improving

the skill score by 22%.

## 1 Introduction

Seasonal snow is characterized by a strong spatial and temporal variability (Mott et al., 2018) arising from several processes such as preferential deposition, wind transport, differential radiation and heat fluxes, metamorphism. This is a challenge for spatially-distributed modelling efforts as point measurements have a limited utility. Manual snow measurements (of depth,

snow water equivalent (SWE) or density) are present in populated areas, but this is not the case for remote mountain ranges (Orsolini et al., 2019), or above the treeline (Treichler and Kääb, 2017). Similarly, field campaigns are expensive and limited in their spatio-temporal extent. Remote sensing is thus the most promising method to estimate the spatial distribution of snow



(Clark et al., 2011). Moreover, Dozier et al. (2016) state that the study of snow water equivalent (SWE) distribution is the sector in hydrology that would benefit the most from remote sensing innovations.

Optical satellite platforms are routinely used to retrieve the fractional snow-covered area (fSCA) which is the key observed variable to reconstruct the seasonal snow evolution (Margulis et al., 2016). Several products are available, including operational ones such as long-term but coarse global ESA Snow_cci snow cover data from AVHRR (Naegeli et al., 2022), moderate resolution global snow cover data from MODIS and VIIRS (Riggs et al., 2017), and higher resolution snow cover retrievals from Sentinel-2 and Landsat available in limited regions (Gascoin et al., 2019). The accuracy of fSCA retrievals varies considerably

depending on the retrieval algorithm and sensor employed (Aalstad et al., 2020). Both coarser and higher resolution fSCA retrievals have been shown to help reconstruct the evolution of the seasonal snowpack across entire mountain ranges (Alonso-González et al., 2021; Margulis et al., 2016).

Accurately measuring snow amounts (i.e., mass or depth) from space is still a major scientific challenge (Dozier et al., 2016), but many approaches are currently used. Satellite measurements with passive microwave sensors have coarse spatial resolution

(tens of kilometers) and saturate with a deep snowpack, limiting their applicability in complex terrain,(Foster et al., 2005) as in the state-of-the-art global passive microwave SWE product GlobSnow v3.0 (Luojus et al., 2021) where mountain regions are masked out. Active microwave sensors have also been experimented with, and Lievens et al. (2022) obtained snow depth estimates at medium spatial resolution (500 m) from Sentinel-1 Synthetic Aperture Radar (SAR) backscatter information and an empirical change detection algorithm.

In the last decades, unpiloted aerial vehicles (UAV) and airplanes have been widely used to map the spatial distributions of snow depth on a catchment to smaller scales. Photogrammetry (Eberhard et al., 2021), and light detection and ranging (lidar; Geissler et al., 2023; Harder et al., 2019) have been extensively used for this purpose. The most notable effort is the airborne snow observatory (ASO; Painter et al., 2016)), where snow depth has been extensively mapped over a large number of basins over the western part of the North American continent. However, the costs of performing such campaigns remain high and

organizationally complex, making this approach prohibitively expensive for mapping seasonal snow globally.

Photogrammetry can be applied also with space-borne platforms acquiring high-resolution stereo imagery (Marti et al., 2016)), but the costs of acquiring imagery from private companies that operate very high-resolution optical satellites with the required specifications currently limit its use. Lidar technology is also used onboard satellites, for example snow depth has been estimated with the NASA satellite ICESat (Treichler and Kääb, 2017). However, the coarse footprint limits the applications of this

technology in complex topography.

In October 2018, the ICESat-2 mission was launched. The Advanced Topographic Laser Altimeter System (ATLAS) mounted onboard has significantly better sensor characteristics for measuring snow depth. It emits 532 nm (green) laser pulses that illuminate six parallel profiles along which it measures the earth's surface elevation (Markus et al., 2017). On average, for an area located at mid-latitudes extending around 10 km in longitude, ICESat-2 is expected, in the absence of clouds, to

scan the surface with the 6 beams once or twice during the snow season. The geolocated photons have a centimetric vertical



measurement error (although on flat terrain Markus et al., 2017) and the geolocation horizontal accuracy is estimated at 3 to 4 m (Magruder et al., 2021). ICESat-2 data products derived from the spatial aggregation of photons over tens to a hundred meters have been used to measure snow depth by differencing with a digital elevation model (DEM) acquired during snow-free conditions. From comparison to airborne lidar snow depth observations, Enderlin et al. (2022) found that the median absolute

deviation (MAD), an index of accuracy, is around 0.2 m for slopes $< 5°$, while it increases to be $> 1$ m for slopes $> 20°$. Deschamps-Berger et al. (2023) found similar results: the random error (precision) was 0.5 m for $< 10°$ sloped terrain. Besso et al. (2024) improved the results in the same basins of the two works by focusing on customized data products generated with the SlideRule Earth service (Shean et al., 2023).

All of the aforementioned observations have limited value in directly estimating snow water resources, because they are only

able to capture a subset of the full state of the snowpack at one to several points in time and space (because of their coverage, and spatial and temporal resolution). In order to capture the spatiotemporal evolution of the snowpack, these observations have been used to constrain a plethora of snow models with varying complexity, from simple empirical models with one to several parameters (Hock, 1999), to physically-based models that represent the snow energy and mass balance with many distinct layers (Lehning et al., 2002). Herein, we focus on an intermediate complexity snow model, namely the Flexible Snow Model

(FSM2 Essery, 2015), which is a compromise between a detailed representation of physical processes that influence the key snow hydrological state variables and parsimonious parametrizations that allow for increased computational efficiency. The main limitation of all snow hydrology models, independently of complexity, has been shown to be the atmospheric forcing data (Raleigh et al., 2015), especially in the spatially-distributed case when the forcing needs to be extracted from larger scale atmospheric model outputs such as coarse resolution (30 km) global atmospheric reanalyses (e.g. ERA5; Hersbach et al.,

75  2020).

Data Assimilation (DA) enables the fusion of noisy observations with uncertain models in a Bayesian statistical framework (Evensen et al., 2022), obtaining (statistically) optimal estimates with an associated uncertainty. This technique, especially ensemble-based (Monte Carlo) implementations, has shown considerable promise in the snow science community to meet the reconstruction and forecasting requirements needed to more accurately map the water storage services that snow provides to

downstream ecosystems and communities (Girotto et al., 2020). In terms of time dynamics, these schemes can be employed either in a strictly sequential forward manner as filters (e.g. for initializing short-term snow hydrological forecasts Mott et al., 2023), or instead as retrospective smoothers that allow information from observations to transfer backward in time, yielding a constrained and consistent reconstruction (ideal for snow reanalysis problems Margulis et al., 2016).

Ensemble DA methods can be further subdivided between particle and Kalman methods (Evensen et al., 2022). Particle filters

(Leisenring and Moradkhani, 2011) and smoothers (Margulis et al., 2015) are particularly popular snow DA methods given their flexibility, ease of implementation, and relative lack of assumptions. However, particle methods are prone to undesirable ensemble collapse due to weight and path degeneracy (Murphy, 2023), especially in higher dimensional spatio-temporal problems (Cressie, 2011) that remains an active area at the frontier of particle DA research (Evensen et al., 2022).



Herein, we restrict our attention to ensemble Kalman methods (Evensen et al., 2022) which, despite (and thanks to) stronger
Gaussian linear assumptions, have been shown to be robust also in very high-dimensional geophysical DA problems (Carrassi
et al., 2018).These ensemble Kalman techniques represent distributions through an ensemble of model realizations that are
updated in state and/or parameter space using available observations. Moreover, the use of iterative ensemble Kalman updates
that temper the likelihood such as the ensemble smoother with multiple data assimilation scheme (ES-MDA Emerick and
Reynolds, 2013), and the deterministic version (DES-MDA Alonso-González et al., 2023) has been shown to strongly mitigate
the negative impact of a linear forward model assumption implicit of such methods (Aalstad et al., 2018; Evensen et al.,
2022).

Ensemble Kalman methods have been widely applied to various snow DA problems and have been used to assimilate SWE data
from stations(Magnusson et al., 2014) and passive microwave satellite retrievals (De Lannoy et al., 2012), snow depth data from
stations (Stigter et al., 2017) and drones (Alonso-González et al., 2022), and fSCA satellite retrievals (De Lannoy et al., 2012;
Stigter et al., 2017). Moreover, the ensemble smoother, a batch ensemble Kalman smoother (see Evensen et al., 2022; Alonso-
González et al., 2022), was suggested as a method for Bayesian snow reconstruction by Durand et al. (2008) and has been
subsequently used to assimilate both moderate (Oaida et al., 2019) and higher resolution (Girotto et al., 2014) fSCA retrievals.
More recently, iterative ensemble smoothers based on the ES-MDA have shown promise in the assimilation of satellite-based
fSCA retrievals (Aalstad et al., 2018) or drone-based snow depth retrievals (Alonso-González et al., 2023).

Most snow DA research, with a few exceptions (e.g. Magnusson et al., 2014; De Lannoy et al., 2012; Cho et al., 2023), has
focused on purely temporal DA where the snow in each model grid cell (or more generally spatial unit) is simulated and
updated independently of its neighboring cells and the observations therein. However, De Lannoy et al. (2022) recommend a
greater adoption of spatio-temporal multivariate DA. Recent studies have demonstrated the added value of spatio-temporal DA
in exploiting spatially sparse snow depth observations propagating the information through covariances based on geographical
distance (Cluzet et al., 2022). Alonso-González et al. (2023), in contrast with the other aforementioned exceptions, have
shown promising results with the ES-MDA scheme and a prior covariance matrix dependent on pixel proximity in a multi-
dimensional space based on morphometric terrain features. However, this approach has yet to be extended to emerging yet
sparse space-borne observations (see comment 6 ?) of snow depth, such as profiles derived from the laser altimeter on ICESat-
2(Deschamps-Berger et al., 2023; Besso et al., 2024).

In this study, we aim to demonstrate for the first time the value of assimilating snow depth retrieved from the satellite laser
altimeter ICESat-2 in a small experimental catchment. While other studies have focused on evaluating spatially aggregated
data products from this satellite altimeter (Deschamps-Berger et al., 2023; Enderlin et al., 2022; Besso et al., 2024), we use
the geolocated photon data product ATL03 (Neumann et al., 2019) with finer spatial resolution. The validation of this product
is undergoing work, but preliminary results show good agreement in comparison to drone-based snow depth maps. Because
of the temporal and spatial sparsity of these observations, their utility for direct mapping of snow depth via temporal DA is
limited. Hence, we showcase the use of ICESat-2 data in a spatio-temporal DA scheme (Alonso-González et al., 2023) that
enriches this data in that it can now be used to constrain distributed seasonal snow models for an entire catchment.



We propose the joint assimilation of ICESat-2 snow depth and fSCA from Sentinel-2. This is compared to assimilating only ICESat-2 data or fSCA data alone, whereby the latter is widely available and routinely used in the snow DA community. The

two datasets have complementary features: ICESat-2 retrieves snow depth directly, but only along profiles; while fSCA has an indirect correlation with snow depth, but this dataset is spatially distributed. Our hypothesis is that the joint assimilation will be able to exploit these to better infer the seasonal snow evolution.

## 2 Study Area and Data

The study area is the Izas experimental catchment (Revuelto et al., 2017), located in the Central Spanish Pyrenees. The elevation
ranges between 2075 and 2325 m a.s.l. and the total annual precipitation typically sums up to 2000 mm, with around half of this in the form of snowfall as is typical for such a sub-alpine environment. Snow usually covers a large fraction of the catchment from late November to the end of May. The vegetation type is a high mountain steppe: most of the surface is covered by bunch grass. We chose this site because of the availability of several spatially distributed drone-based snow depth measurements (Alonso-González, 2022) in the 55 ha area highlighted with black in Figure 1 (Revuelto et al., 2021). The drone surveys were
conducted in 2020 on 14/01, 03/02, 24/02, 11/03, 29/04, 3/05, 12/05, 19/05, 26/05, 02/06, 10/06, 21/07 and were already used in various DA experiments (e.g. Alonso-González et al., 2022, 2023). The original resolution is 1 m, and the measurement error is assumed to have a standard deviation equal to 20 cm. The right panel of Figure 1 shows the spatial distribution of snow depth close to the seasonal peak. Very high snow depths ($> 300$ cm) are accumulated under the ridgeline on the west side of the area, in the deep valleys throughout the catchment, and at the foot of the slope on the east side. Very low snow depths
($< 50$ cm) can be seen in the south-southeast facing aspects located on the north side of the catchment.

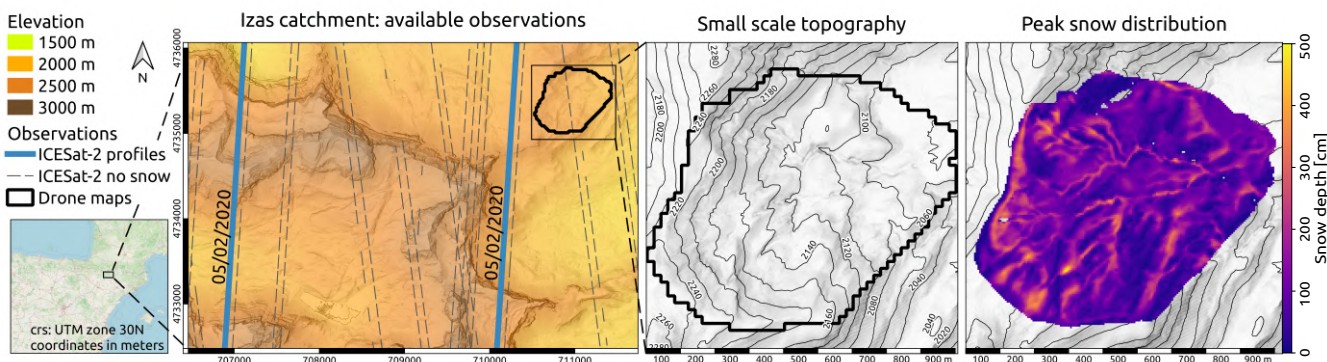

**Figure 1.** Topography of the Izas experimental basin, located in the Pyrenees. The raw DEM data is obtained from the Centro Nacional de Información Geográfica. Blue lines: ICESat-2 snow depth profiles in 2020 used in the experiments; gray dashed lines: ICESat-2 snow-free profiles that are used for coregistration purposes; black area: catchment where 12 snow depth maps were acquired during the 2020 snow season. The small-scale topography, shown in the middle panel, dictates the peak snow depth patterns shown in the right panel, which was acquired on 11th of March at 1 m resolution and is obtained from Alonso-González (2022).



We obtain snow depth by subtracting the snow surface elevation measured by the ICESat-2 satellite altimeter to a snow-off DEM. The ATLAS instrument onboard the National Aeronautics and Space Administration (NASA) satellite ICESat-2 provides very accurate surface height measurements along six beams (Markus et al., 2017). We use the level 2A product ATL03 (Neumann et al., 2023), which we downloaded through the SlideRule python-client (Shean et al., 2023). This low-level

product combines the time of flight data with the location of the satellite to obtain the geolocation of single photons' reflection events. The ATLAS transmitter generates six beams (across-track direction). The beams are arranged in three pairs, with a strong and weak beam located at a distance of 90 m, while the pairs are spaced 3.3 km. The signal ratio between strong and weak beams is 4 to 1. One pulse of the 532 nm laser illuminates a region ca. 14 m in diameter. Pulses are transmitted every 1.5 ns, hence the footprints are spaced 0.7 m in the along-track direction and have a large overlap. The expected number of

photons backscattered to the ATLAS receiver for a single strong beam shot from a reflective surface such as snow ranges from 7 to 3 depending on the slope and roughness of the surface (Neumann et al., 2019). We select two of the strong beam profiles acquired on the 5th of February 2020, as they sample snow depth during the accumulation phase. We discard the third strong beam as it is located further away at lower elevations and in forested terrain, and the weak beams as preliminary results showed their signal-to-noise ratio is too weak for this application. ICESat-2's next acquisition during the 2020 snow season in the study

area is in May, but the cloudy conditions made these profiles unreliable. ICESat-2 measures the surface elevation through the geolocation of photon reflection events, and we access this information through the low-level ATL03 data product (Luthcke, 2021). Most of these events happen on the highly reflective snow surface during the cold season, but there is a substantial amount of noise due to solar radiation or double bounces (Neumann et al., 2023). Photon events are divided into several classes depending on the identified location of the event (e.g. ground, canopy, top of the canopy) in the algorithm vegetation product

ATL08 (Shean et al., 2023).

The DEM used as the snow-off reference surface is available thanks to the Spanish government Plan Nacional de Ortofotografía Aérea (PNOA) lidar initiative and its spatial resolution is 2 m. The accuracy in terms of RMSE for this lidar-based model is declared to be 20 cm in the vertical direction and 30 cm in the horizontal plane (Centro Nacional de Información Geográfica). Moreover, as we will detail in the following section 3, we use snow-off ICESat-2 profiles to evaluate the vertical offset between

the mentioned DEM and the ICESat-2 acquisitions. We employ 18 profiles depicted with a gray dashed line in Figure 1.

In addition to the ICESat-2's snow depth data, we also employ snow cover information from high-resolution ($\sim 10$ m) multispectral satellite imagery. We used surface reflectances (the Level 2A product) obtained from the MultiSpectral Instrument onboard the Sentinel-2A and Sentinel-2B twin satellites (Drusch et al., 2012) operated by the European Space Agency as part of the Copernicus Programme. The Sentinel-2 imagery was downloaded from Google Earth Engine (Gorelick et al., 2017),

which is a cloud-based platform that harvests open Earth observation data from its original source, in this case Copernicus. By manually selecting cloud-free imagery, we obtained a total of 19 scenes covering the entire study area and snowmelt season. The acquisition dates of the used scenes are shown with blue stars in panel a) of Figure 4, and are irregularly spaced between the 5[th] of February (before peak snow) and the 17[th] of July 2022 (complete melt-out), with a median (maximum) spacing of 5 (33) days.



## 3 Methods

### 3.1 ICESat-2 snow depth retrieval

As mentioned in the previous section, the laser altimeter ATLAS on-board of the satellite ICESat-2 geo-locates photon reflection events to retrieve the surface elevation. We filter such events by selecting only the ones classified as ground, as results from Besso et al. (2024) indicate such filtering improves the median absolute error. Moreover, we assign the photon events a weight based on the local neighborhood density, using the Yet Another Photon Classifier (YAPC) algorithm (Sutterley and Gibbons, 2021). This determines the significance of individual photon events with a customized inverse-distance weighted kNN algorithm. The neighborhood is defined with a window length (parallel to the line of flight) of 5 m and a 3 m height. The rationale behind this is that photons returning from the ground have a large number of neighbors as it is less likely for photons reflected from atmospheric particles or objects above the ground to be clustered together. For each $\sim 4$ km profile we select $60\%$ of the photons with the largest significance according to YAPC. In Figure 2, the orange photons' size is proportional to their YAPC score, while the filtered-out photons are grey. Before comparing the ATL03 photon events to the snow-free reference surface elevation it is necessary to co-register this dataset with the snow-off DEM. Every beam is independently co-registered with a horizontal displacement, and a vertical offset common to all the acquisitions and beams is obtained by computing the median of all the snow-off acquisitions vertical offsets, as other studies have done (Enderlin et al., 2022; Besso et al., 2024). We employ the Nuth-Kääb algorithm to obtain the horizontal shifts (Nuth and Kääb, 2011), implemented in the xdem python library (Dehecq et al., 2021).

Snow depth is computed for each selected photon event by subtracting the elevation from the co-registered DEM. We linearly interpolate the DEM to obtain the snow-off elevation at the location of the photon event. Subsequently, we divide the snow depth observations into cells with a 20 m spatial resolution, in order to match the spatial resolution of the simulation (see 3.6). Since the ICESat-2 orbit is $92°$, each cell has around 29 footprints summing up to $45\pm18$ photons after the YAPC filtering. As some of the cells defined by modelling grid might have very few measurements, we filter out cells where less than 10 photon reflection events. In addition, cells with an average slope larger than $40°$, as the horizontal positioning uncertainty makes snow depth retrievals not reliable for steep terrain. In Figure 2, a 400 m transect with a comparison between the co-registered photons and the high-resolution DEM is shown (upper panel) as well as the snow depth sample distribution available along the transect (lower panel). We sample the obtained snow depth distribution to retrieve an observation in each cell with the median operator. To estimate a domain-consistent statistic for the spread of the snow depth observation error ($\sigma_y$), we compute the standard deviation of snow depth samples in each cell and average it over the profiles, obtaining $\sigma_y = 0.92$ m.

### 3.2 Sentinel-2 fSCA retrieval

The fSCA retrieval algorithm is based on surface reflectances estimated from MSI onboard the Sentinel-2 satellites in 6 bands in the visible, near-infrared, and shortwave infrared. By comparing the reflectances measured in these bands to modeled spectra for snow and snow-free endmembers we infer the fSCA using spectral unmixing via a fully constrained least squares algorithm.



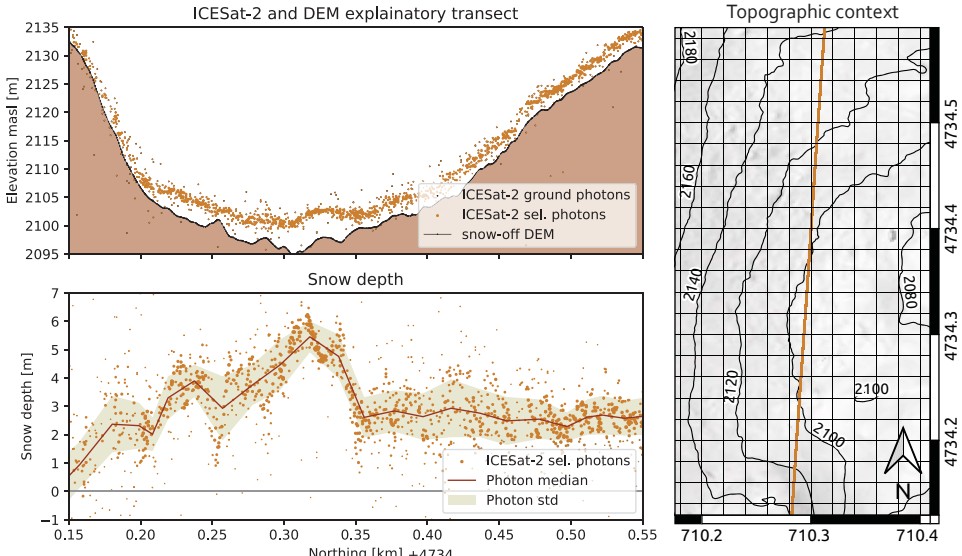

**Figure 2.** Left panel: the topography of the ICESat-2 profile. Upper panel: A 400 m long segment where the photon events selected are compared with the snow-off information obtained from the 2 m resolution DEM. Lower panel: the median operator is applied to snow depth observations from the selected photon on a 20 m moving window and the shaded area represents the 2 standard deviation range around the median.

As shown in Aalstad et al. (2020), this approach can outperform simpler linear regression and thresholding-based approaches to fSCA retrieval, albeit at a higher computational cost. This unmixing approach to retrieving fSCA from Sentinel-2 imagery has been successfully applied both in the high-Arctic (Aalstad et al., 2020), as well as at Alpine sites (Pirk et al., 2023) in
Norway. The retrievals were performed in the native 10 m grid of the visible and near-infrared reflectances from Sentinel-2, and were subsequently regridded to the 20 m spatial resolution of the simulations 3.6 through averaging and subsequent inverse distance weighting. We estimate the observation error for the fSCA retrievals at 20 m resolution to be $\sigma = 0.34$. As independent validation estimated the observation error $\sigma_N$ at 100 m resolution to be equal to $\sigma_N = 0.07$ (see Table 2 Aalstad et al., 2020), we expect the error at coarser resolution to improve according to the central limit theorem $\sigma_N = \frac{\sigma}{\sqrt{N}}$, where $N$ is the number of
independent 20 m cells being aggregated in the coarser validation ($N = 25$ in this case). Thus, the disaggregated observation error at 20 m resolution for these fSCA retrievals should be on the order $\sigma = \sigma_N \sqrt{N} \simeq 0.34$ from which we obtained our estimate.





## 3.3 Modelling

We simulate the snowpack at 20 m spatial resolution with the Flexible Snow Model (FSM2 Essery, 2015, 2023). This
intermediate-complexity model represents the snowpack with up to three different layers and solves the coupled mass and
energy balance equations to simulate the seasonal evolution of snow. To obtain a more comprehensive snowpack representa-
tion, 7 physical processes are parameterized with the most detailed process representation among those available in the snow
model. These parametrizations are: albedo decay with elapsed time since the last significant snowfall, thermal conductivity
depending on snow density, density influenced by overburden and metamorphism, turbulent fluxes diagnosed using the Monin-
Obukhov similarity theory, and melt-water percolation depending on gravitational drainage, fractional snow cover asymptotic
to snow depth.

We drive the simulations with meteorological forcing derived from the ERA5 reanalysis (Hersbach et al., 2020). Because the
coarse resolution of 30 km misses the subgrid topography-driven heterogeneity of the atmospheric variables, we downscale the
reanalysis with TopoSCALE (Filhol et al., 2023). This process uses the pressure level data to interpolate the forcing variables
at the cell elevation, and radiation components are scaled depending on the topography. The TopoSUB routine in TopoSCALE
allows for efficient semi-distributed spatial downscaling through topography-based clustering (Fiddes et al., 2019). We select
400 as an appropriate number of clusters to run the semi-distributed downscaling for this rather small area (equal to the extent
of the map in the left panel of Figure 1). The obtained semi-distributed forcing is then mapped back to the 20 m fully distributed
grid. Such a combination of topographic downscaling (although another downscaling model was used) and the snow model
235 FSM2 has already been used in this area in Alonso-González et al. (2022, 2023). Note that to save computational resources we
simulate only the cells in the domain where an ICESat-2 observation or the drone validation is available ($\sim$ 1900 cells).

## 3.4 Data assimilation

The Multiple Snow DA System (MuSA Alonso-González et al., 2022) allows the execution of various forms of ensemble-based
snow DA. Therein, the prior uncertainty is represented by the spread of the ensemble members. Each member is an FSM2 sim-
240 ulation obtained by perturbing a selection of forcing variables. In the presented experiments, the perturbed forcing variables are
air temperature, precipitation and downwelling longwave radiation. The perturbation parameters are time-invariant throughout
the water year, and the prior perturbation parameters are extracted via transformations from a logit-normal distribution rather
than Gaussian, to restrict the perturbation within defined bounds (Aalstad et al., 2018; Guidicelli et al., 2023). The nature
of the perturbation is multiplicative for the precipitation (to prevent non-physical negative values) and additive for the other
245 variables.

The ensemble members representing the prior perturbation parameter distribution are updated with a DES-MDA with four
iterations, as the mapping from perturbation parameters to observations – the snow model FSM2 – is clearly non-linear.
We select 40 as an appropriate number of ensemble members in order to adequately represent the prior distributions while
maintaining a reasonable computational cost.





## 3.5 Spatial propagation of information

The key to spatially propagate information from local observations in space is in the construction of the prior covariance matrix with spatial dependence (Cressie, 2011). Details on the practical implementation (and the theoretical background) of the system can be found in Alonso-González et al. (2023). As the spatial distribution of snow depth is strongly governed by topography, and as the relative patterns are often repeated year after year, we follow a concept introduced in experiment III of Alonso-González et al. (2023). We employ a generalized prior correlation function dependent on the proximity of cells in a feature space. We selected the following features as the dimensions for the generalized space:

1. **TPI**: the topographical position index (Weiss, 2001) is computed as the difference of the cell's elevation compared with its neighborhood defined as a 24 m radius. It represents the exposure of the terrain at the mentioned scale, so that cells with negative TPI are located in a concavity or a valley and positive TPI are cells in convex terrain such as a ridge relative to its surroundings;

2. **Sx**: the maximum upwind slope parameter of Winstral et al. (2002) provides information about topographical sheltering of individual cells. This index is computed in the prevailing wind direction: northwest for the Izas study site. It corresponds to the maximum elevation gradient between the selected cell and all the cells upwind that lie within a maximum distance of 200 m ;

3. **CSMD**: the Climatology of Snow Melt-out Date is obtained by averaging the date (day-of-water-year) when the snow melted out in a selected pixel, extracted from Sentinel-2 fSCA time series provided by the Theia land data center (Gascoin et al., 2019). Five water years were used (2017-2021).

The topographic features (points 1 and 2 of the list) are computed with the high-resolution snow-off 2 m DEM to capture small-scale topographic effects. To match the simulation resolution of 20 m, we use the average of all high-resolution values within each model grid cell. The terrain indexes, their size and direction were selected based on the results of regression experiments targeting snow depth in the Izas experimental catchment (Revuelto et al., 2014). Figure 3 shows the spatial distribution of the aforementioned features at the modelled resolution (20 m) in the experimental catchment as well as a scatter-plot showing the relation between snow depth and the selected coordinates and snow depth.

As the dimensions have different units, each is standardized by applying a z-score, so that the set of coordinates has a null mean and standard deviation equal to 1. Secondly, CSMD coordinates are increased (multiplied by 1.5) to make their relative weight larger as this feature correlates the most with snow depth. This effectively creates a space where we can measure similarity between cells. Covariance localization is the practice of limiting the effect of long-range spurious correlation (Evensen et al., 2022). We employ the widely used damping operator in the Gaspari and Cohn correlation function (GC; Gaspari and Cohn, 1999), which is defined by a single hyper-parameter: the correlation length scale. In practice, we define the neighborhood of each cell as the set of cells that are located in the feature space within two times the correlation length scale. The closer the cells the more similar the snowpack the system will constrain the cells to be, and the further away the less strict the system will



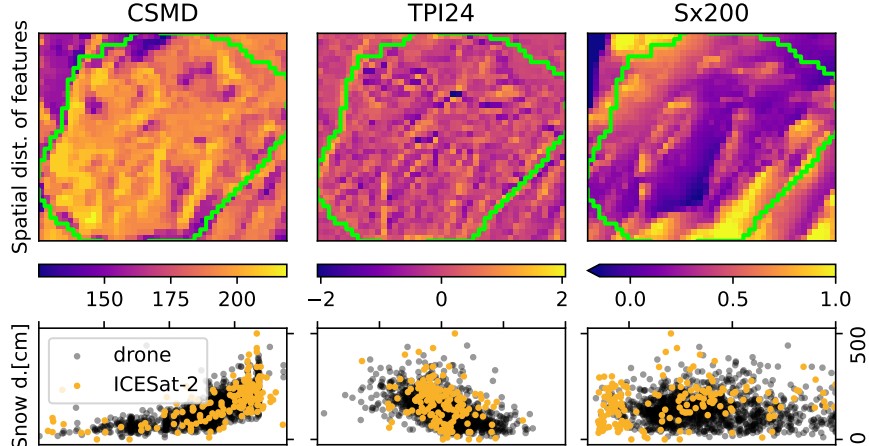

**Figure 3.** Upper panels: spatial distribution of the three feature dimensions (Climatology of Snow Melt-out Date: CSMD, Topographical Position index: TPI, Winstral index: Sx) in the experimental catchment, outlined with a green line. Lower panels: the dimension's relation with snow depth from the drone map acquired on the $3^{rd}$ of February (black) and with ICESat-2 for the observed profiles on the $5^{th}$ of February (yellow).

enforce similarity. Cells outside each other's neighborhood will not influence the respective simulation. The correlation length scale was set to 1.5 after testing several values and making sure the size of the resulting neighborhoods was acceptable.

## 3.6   Experiments

Three experiments are carried out where different observations are assimilated for the water year 2020. Only information from the ICESat-2 snow depth retrievals – located outside the measured drone field as visible in Figure 1 – is spatially propagated, as fSCA maps are available for all cells in the model domain. To allow a comparison where we change exclusively the assimilated variables, we use the same spatially correlated prior for all the experiments. The three experiments are designed as follows:

– Snow **c**over experiment (**C**): Temporal assimilation of local fSCA retrievals from Sentinel-2. For each grid cell, all the available local (not neighboring) fSCA observations are assimilated. We use this as a baseline simulation.

– Snow **d**epth experiment (**D**): Spatio-temporal assimilation of snow depth retrievals from the ICESat-2 satellite altimeter. The observations of the snowpack on the $5^{th}$ of February are assimilated. Since the profiles are located outside the experimental catchment, the information is spatially propagated using the dimensions we introduced above and depicted
in Figure 3.

– **J**oint assimilation experiment (**J**): Spatio-temporal assimilation of local fSCA and neighboring snow depth. All the observations of experiments (**C**) and (**D**) are assimilated, and their respective observation error standard deviations are



the same as for these experiments. Note that only the sparse snow depth observations are spatially propagated, while the spatially-complete fSCA observations are not.

All the experiments are set up with the MuSA system (Alonso-González et al., 2023). The system was already set up for jointly assimilating different observed variables, although joint assimilation was tested only in a temporal DA setting. Each cell is updated with the local observation (if any), located in the same cell position, together with the observations in its neighborhood defined as the set of cells located inside a search radius in the afore-mentioned feature space with dimensions TPI, Sx and CSMD (3.5). This domain localization step greatly reduces the computational effort, as it avoids searching through all the

simulated cells. For this study, we updated the MuSA system in order to select a subset of the available observation types to spatially propagate Alonso-González et al. (2024). This allowed us to exclude fSCA observations (which are spatially complete) from the spatial propagation, which leads to an additional marked decrease in the computational cost.

## 3.7 Evaluation

We use the drone-based snow depth maps of the Izas experimental catchment (Revuelto et al., 2021) as ground truth to in-
dependently evaluate the relative performance of the DA in the three experiments. Their measurement error is typically one order of magnitude lower than the uncertainty in the snow-pack reconstruction, as we resample the snow depth maps to the modelling resolution (from 1 m to 20 m) with the averaging operator. We aim at evaluating the sampled posterior distribution in terms of the resulting simulated snow depth goodness of fit. We evaluate our results in three ways: i) in time, ii) in space and in terms of snow depth distribution and iii) for the entire spatio-temporal ensemble distribution. For i), all the ensemble
member snow depth simulations are spatially averaged over the measured experimental catchment for each day in the simulated water year, and compared to the corresponding spatially averaged drone observations. This way, we can also evaluate how well the experiments perform in estimating the total snow in the catchment. For ii), we visually compare the spatial distribution of the simulations against the drone-based map measured on the 11th of March, the closest acquisition to the seasonal SWE maximum. Since the result of the DA problem is a spatially correlated ensemble representing a statistical distribution, we show
one single ensemble member simulation in order to appreciate the spatial structure embedded in the simulation. We spatially average the ensemble members and pick the member whose average snow depth state is selected by the median operator for the 11th of March. To evaluate the inference results of the three experiments (iii), we employ the Continuous Ranked Probability Score (CRPS Hersbach, 2000), as this metric evaluates the performance of the inferred distribution represented by all the ensemble members (rather than a single point-estimate such as the median of the distribution) of each simulated cell in terms
of snow depth compared to the observed reference. In particular, the CRPS quantifies both the precision (certainty or confidence) and the accuracy (ability to match the observations) of the ensemble as a whole. This is a strictly positive score, where a perfect match between the compared distributions would result in a score of 0, while the larger the CRPS score the worse the result. We compute the CRPS metric for each experiment for all the available drone-based snow depth maps. To evaluate how the experiment's performance varies in time we average the experiment's score throughout the measured catchment; while to





evaluate the spatial distribution of the errors we present two maps per experiment with the average score for accumulation and melting season.

## 4 Results

Figure 4 summarizes the results for Experiment **(C)**. The time series in panel a) shows the prior and posterior ensemble members averaged over the Izas catchment. The ensemble of simulations is precise (low ensemble spread) and accurate (good
match to validation data) towards the end of the snow season. However, the ensemble spread is larger in the snow accumulation months and the ensemble median overestimates the catchment-average snow depth. From the map in panel b), it is discernible that the simulation correctly reconstructs the observed snow depth patterns in a relative sense (panel c)): the areas with larger-than-average snow depth are correctly recognized, as well as the ones with lower-than-average snow depth, despite the absolute values not being correct. The histograms in panels d) and e) highlight that the simulation average overestimates snow depth at
peak SWE by $47\%$. In the simulation, only $10\%$ of the simulated cells are inferred to have less than $150$ cm of snow, while in the drone validation about $50\%$ of them are measured in this part of the range.

Figure 5 summarizes the results of Experiment **(D)**. Panel a) shows that the assimilation of the sparse snow depth profile data from a single date substantially narrows and improves the posterior distribution compared to the prior. Notably, during the winter season, the accuracy of the posterior is improved (the ensemble median and the black validation points are closer)
compared to the first experiment **(C)**, and also the precision is improved (the ensemble spread is lower). Towards the other end of the season, both accuracy and precision largely decrease compared to the first experiment, despite still improving from the prior simulation. Panel b) shows that the posterior simulation is able to match only some of the spatial patterns that the drone map shows in panel c), such as in the northeast of the catchment. Also, the valley affected by wind drift and the corresponding wind-blown ridges in the southeast portion of the catchment are partially recognizable. However, patterns in the high-elevation
part of the catchment (west side) are not reproduced. The histograms in panels d) and e) show that the simulated snow depths at peak-SWE have similar ranges, despite the mean being over-estimated by $33\%$. In contrast with Experiment **(C)**, low snow depth areas are represented as in the measured map. However, also in this experiment, $32\%$ are simulated with very high snow depth ($> 300$ cm) despite the fact that in the drone validation only $9\%$ of the cells are measured with such snow depths.

Figure 6 summarizes the results of Experiment **(J)**, where we use all the available observations. The time series in panel a)
shows that the inference produced a precise result (low spread) throughout the water year. In the accumulation period and up to the peak-SWE, the catchment-average snow depth is accurately reconstructed. Notably, for the 11th of March acquisition the measured and simulated average snow depth differs only 2 cm. However, there is a negative bias in spring, which we do not see in experiments **(C)** or **(D)**. The comparison of simulated and measured maps in panels b) and c) shows that the posterior simulation is able to match only some of the relative spatial patterns. For example, the accumulation below the ridge
on the western border of the catchment is simulated with higher-than-average snow depth only in its center and south portion of the feature, and not in its northern part. Only some of the deep valleys and depressions in the south and some of the flat

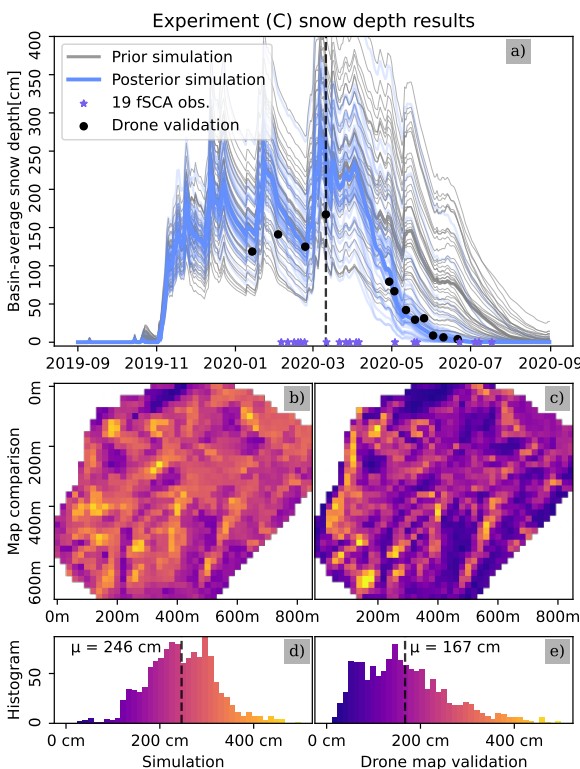

**Figure 4.** Results from experiment **(C)**. Panel a): prior (gray) and posterior ensemble simulations (blue) as catchment-average snow depths over the whole water year. The black points are the drone-based snow map averages serving as validation data. Panel b): simulated snow depth map for 11th March 2020 (date shown with a vertical dashed line in panel a): the median ensemble with respect to the catchment-average peak SWE is selected, color map explained in panels d)/e). Panel c): corresponding drone-based snow map at the model's spatial resolution (20 m). Panels d) and e): snow depth histograms of the maps in the panels above.

accumulation areas in the north-east have the correct relative snow depth. The nearly snow-free south-facing slope in the north is correctly simulated. In terms of distribution, the histograms in panels d) and e) show clearly that the simulation reproduces the mean and the frequencies of the tails. Low snow depths (<150 cm), simulated for $47\%$ of the cells, match the number of cells measured with such range ($46\%$); as well as for very high snow accumulation values ($> 300$ cm): $5\%$ of the cells are simulated and $9\%$ measured.

Scoring the three experiments' whole ensemble allows for a more quantitative comparison of the inference, as panels b) and d) of the previous figures only show one ensemble member's simulation. Figure 7 shows the CRPS computed against the drone acquisitions for the available snow depth maps, averaged in space (left panel), and time (right panels). Focusing on the



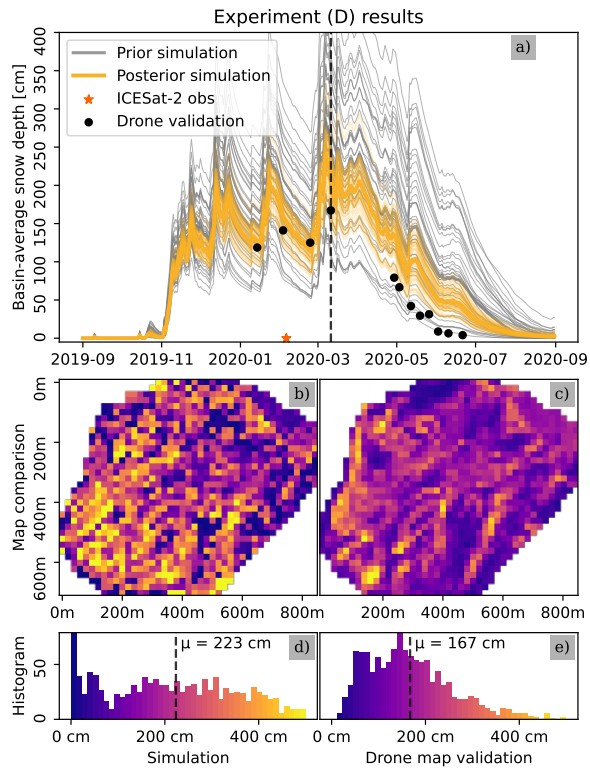

**Figure 5.** Results from Experiments (**D**), presented in the same way as in Figure 4.

right panel, the right panel shows that for all experiments, the largest errors (largest CRPS) are found during the accumulation season, while the errors decrease in the melting season, together with the absolute values of snow depth.

Averaging the CRPSs over the accumulation season, the ICESat-2 assimilation (**D**) shows a similar performance ($48 \pm 12$ cm) to the fSCA assimilation (**C**), whose CRPS is $45 \pm 18$ cm. Both these experiments are substantially worse than the joint assimilation (**J**), scoring $35 \pm 7$ cm. Thus, adding the snow depth profiles to fSCA in the set of assimilated observations lowers (i.e., improves) the error score by $22\%$.

In the melting season, the fSCA experiment (**C**) has very similar results to the joint assimilation (**J**), even being the best for two snow depth maps. The experiment with the ICESat-2 assimilation (**D**) shows the worst performance, with the highest CRPS.

Looking at the spatial distribution of the errors (right panels in Figure 7), experiment (**C**) shows that the CRPS is generally low for both accumulation and melt season. Despite slightly higher values in the accumulation season, only $2\%$ of the cells have a





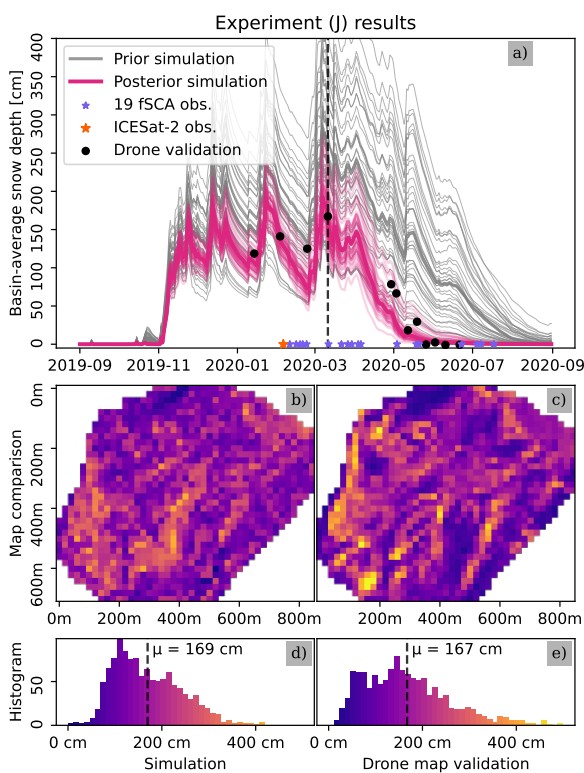

**Figure 6.** Results from Experiments **(J)**, presented in the same way as in Figure 4.

very high CRPS ( > 100 cm). For the accumulation season, the location where large errors cluster are the windblown ridges in the center-south of the experimental catchment where a close look at panels b) and c) of Figure 4 shows that low snow depth is inferred instead of snow absence. Experiment **(D)** performs better during the accumulation season: $10\%$ of the cells have a very high CRPS ( > 100 cm) in the melting season, and these are clustered in the south-west of the catchment. The low elevations

on the eastern side of the catchment have lower scores (compared to higher elevations) in both accumulation and melt season. Experiment **(J)** shows a very similar error pattern distribution in both the accumulation and ablation season (in contrast with the other experiments). Less than $4\%$ of the simulated cells have a very high CRPS (>100 cm) for both accumulation and melting season. An inspection of panels b) and c) of Figure 6, shows that very high errors are located where very high snow depth ( > 300 cm) are measured: the accumulation areas under the ridge on the west side of the border and some of the areas in

the valleys in the south of the catchment.

Focusing on the use of the novel ICESat-2 snow depth retrievals, the experiments show that adding these observations for constraining the inference of the seasonal snow evolution has mostly a positive impact on the simulation results. Comparing the



CRPS over the experimental catchment of experiment **(D)** to the prior (no observation assimilated) shows an improvement of 7.5 cm (3.5 cm) for the accumulation (melting) phase of the season. When adding snow depth to fSCA in the set of observations

– hence comparing experiment **(J)** to **(C)**– the improvement is 10 cm clear for the accumulation part of the season. However, there is a slight decrease of 4 cm in the score for the melting season, despite the left panel of Figure 7 showing experiment **(J)**'s score being worse only for two of the drone acquisitions.

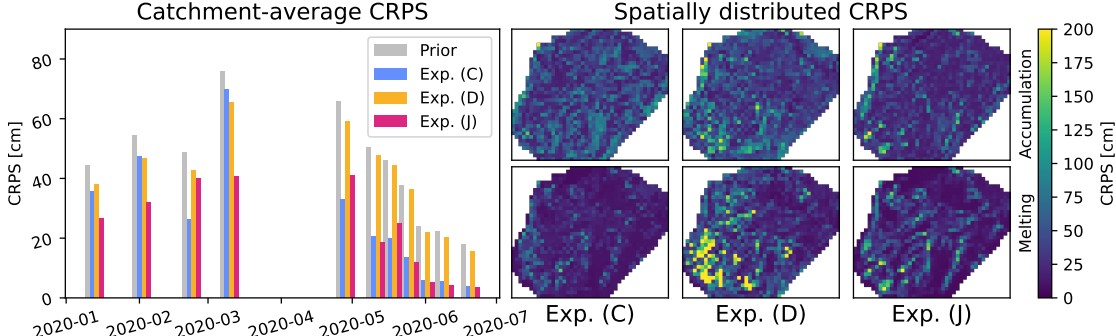

**Figure 7.** The performance of the three assimilation experiments is presented through the Continuous Ranked Probability Score (CRPS), where a perfect match of the compared distribution would score 0, while the larger the score the worse the result. In the left panel, the temporal evolution of the experiments' results is shown by spatially averaging throughout the experimental catchment. In the upper right panels, the spatial distribution of the errors is shown by averaging the CRPS computed for the validation maps acquired during the accumulation season (up to and including the $11^{th}$ of March); while the lower right panels show the same metric averaged over the validation maps acquired in the melting season.

## 5 Discussion

In this paper, three DA experiments are carried out. For all of them, the assimilation algorithm as well as the prior are the same.

We use observations of different snow variables, and all of them successfully improve the prior simulation by constraining the simulations with information coming from the observations, obtaining very different results depending on the assimilated variable. The observations in the three experiments are fSCA retrieved from Sentinel-2 **(C)** as well as the novel snow depth retrievals from the laser altimeter ICESat-2 **(D)**, and finally a joint assimilation experiment using both these retrievals **(J)**. To the best of our knowledge, this is the first example of high-resolution (almost hyper) multivariate snow DA. Moreover, this is

the first study not only to assimilate the ICESat-2 snow depth observations but also to show how these observations acquired along profiles located outside the area of interest can be harnessed through spatial propagation of information. This system successfully updates the snowpack states in the Izas experimental catchment with observations located in its proximity, by exploiting topographical and snow climatological similarities to define a correlation function.

In general, DA consists in combining uncertain information coming from models and observations. In our case, the uncon-

strained model shows an overestimation of the snow depth, especially during the melting season (see gray simulations in panel



a) of Figures 4, 5, 6). The errors in the prior can be attributed to two different sources: the forcing data and the model itself (Raleigh et al., 2015). Errors in the forcing data are to be expected and thus, by design, the forcing formulation of DA allows for the correction of such errors. We emphasize that when carrying out high-resolution, spatially-distributed modelling and using coarse reanalysis such as ERA5 as forcing information (despite the topographic downscaling), one has to expect large biases

given that the high spatial heterogeneity of the drivers of snow accumulation and redistribution processes cannot be explicitly represented in the forward modelling. On the modelling side, one can expect that some of the parameterized processes (e.g. albedo decay or precipitation partitioning) in FSM2 suffer from errors. Here, some of the snow processes can be substantially different and not well captured by FSM2's parametrizations, which were originally implemented on a field site in the European Alps (Essery, 2015). With the given prior in our study, there may be a greater need for information about the melting season

rather than the accumulation season. The correction of the main errors in the prior is accomplished through the assimilation of fSCA, and the DA formulation is able to compensate for forcing errors. This could also explain the better performance of experiments **(C)** and **(J)** compared to **(D)**, in which only observations from the accumulation season were assimilated. However, if snow depth profiles were acquired by ICESat-2 later in the season, the performance during the melting phase might also improve when assimilating only snow depth information.

In experiment **(C)**, we simulate the snowpack and assimilated fSCA to create a baseline. It has been shown that, despite fSCA exhibiting a lower instantaneous correlation with early season snow depth in a deep snowpack compared to the end of the season with a melting snowpack (Girotto et al., 2020), the assimilation of fSCA allows for an accurate reconstruction of peak SWE (Girotto et al., 2014). Indeed, experiment **(C)** shows high accuracy and precision towards the end of the season. As the experiments adopt a smoother approach, such information is also propagated backward in time and the posterior simulation

offers an accurate reconstruction for the peak-SWE: the validation is clearly close to the median ensemble spread, but the reconstruction for this part of the season is less precise compared to experiment **(D)**, and both less precise and accurate for experiment **(J)**. In terms of relative spatial patterns, experiment **(C)** shows the best visual agreement with the validation of drone-based maps, despite overestimating the absolute values.

In experiment **(D)**, we assimilate snow depth observations from ICESat-2 on two profiles, spatially propagating this information

to the experimental catchment. The snow depth was acquired on the 5th of February, about a month before peak SWE. As Figure 5 shows, this information leads to a more precise reconstruction of the catchment-average peak-SWE compared to experiment **(C)**. This demonstrates that the spatio-temporal DA is successful, as the information propagated from observations outside the Izas catchment carries more or at least a similar amount of information compared to the temporal-only information propagation that happens in experiment **(C)**. Compared to experiment **(C)**, this simulation has a better agreement with the observed snow

depth histogram distribution, as the range of the snow depth histograms has a better match (panels d and e), Figure 5). However, the relative spatial patterns of the simulation only partially match those of the validation maps (panels b and c), Figure 5). Since the observations we use in this experiment are not direct measurements in the catchment, this result is in the end not surprising: the similarity measure we define is only partially able to propagate snow depth information properly. Nevertheless, single pixels with extreme values located in the basin might not be similar (in terms of topography and meltout date) to the





ones which are observed by ICESat-2. Towards the end of the season, both the precision and accuracy of the simulations are degraded when compared to experiment **(C)**, and the CRPS score is almost the same as for the prior for this point of the season. Here, the timing of the single assimilated observation is an important factor: most of the information of early-season snow depth observations is related to accumulation processes (precipitation perturbation) rather than melting processes, as found by Guidicelli et al. (2023). The very late melt-out date we obtain with this experiment can mostly be attributed to the fact that also

the prior simulates the snowpack with such a late melt-out date. Margulis et al. (2019) showed that assimilating snow depth observations later in the season, when more consistent ablation processes have taken place, could improve the melting season estimates. If the previous suggested improvements would improve the results of the experiments for this setting, this could be used, in principle, in a forecasting system, as snow depth observations have instantaneous value; while fSCA are more useful in a reanalysis setting. There would be the need to speed-up the ICESat-2 processing for the low-level product, as it is now

usually three months.

In experiment **(J)**, we assimilate all the observations used for the previous two experiments. Note that these two sets of observations complement each other's coverage in time and space. ICESat-2 observations occur in February, when a deep snow pack causes fSCA relation with snow depth to saturate, and hence provides little or no information. However, fSCA is spatially complete and thus nicely complements the sparse, but more direct, snow depth observations of ICESat-2, which are located

along two profiles outside the experimental catchment. We show that executing the joint assimilation leads to higher precision (smaller ensemble spread) throughout the water year. This simulation performs better compared to the experiments **(C)** and **(D)** in terms of the CRPS in the accumulation season. For the melting season, experiment **(J)** has a comparable score to experiment **(C)**, in which the melting season seems already accurately modelled. The time series in panel a of Figures 4 and 6 show that in terms of catchment average snow depth, experiment **(C)** performs slightly better. However, Figure 7 explains the spatial

distribution of the errors and it's clear that for most of the cells both the simulations have a similar score to **(C)**. It is in the large snow drifts that experiment **(J)** is not able to simulate large snow depth.

For both experiments **(D)** and **(J)**, improvements in the spatial distribution are expected in the case of ICESat-2 profiles observing snow in a terrain with more similar characteristics. Moreover, herein the process of spatial information propagation depends on a feature space defined only with few characteristics, that has not been tuned systematically to improve the results,

but that was based on previous studies (Revuelto et al., 2014). Optimizing or tuning the selection of the dimensions in the feature space as well as their relative weight and the correlation length scale, despite being computationally expensive, might lead to better results, as Experiments II and III in Alonso-González et al. (2023) show. We envision future work in which the an optimization process could be fruitful for data-rich vast basins such as the repeated measurements in the western U.S by the ASO (Painter et al., 2016). Despite the optimization or inference of the aforementioned hyper-parameters being foreseen to be

very expensive in terms of computational resources, we acknowledge that it could be performed off-line for a single season, exploiting the repeated patterns of the seasonal snow evolution (Revuelto et al., 2014). In terms of methods, we envision the use of hierarchical data assimilation (Katzfuss et al., 2020) for the statistically-optimal inference of the aforementioned hyper-parameters. In simple terms, this would increase the complexity of the DA system by defining a hyperprior on these





hyperparameters that govern the spatial propagation of information. The assimilation would lead to the optimal inference of
the seasonal snow evolution for the current water year after having inferred the spatial hyperparameters. Moreover, the learned
hyperparameters could conceivably be transferred to other water years without the need for a second round of hierarchical
inference. To alleviate computational bottlenecks, hyperparameter inference could be achieved using simpler snow models that
are now available in MuSA.

A final note is warranted concerning the very high-resolution (almost hyper) of the experiments (20 m). We selected this cell
size to test the ability to measure snow depth at a hill-slope scale with the photon counting technology that ICESat-2's ATLAS
is equipped with. The results are not clear yet in this sense, as validation is not available for the measured track. However,
some of the larger spatial patterns are reproduced, even with this very high resolution, in experiment **(D)**. Lower resolutions are
also adequate for water resources mapping, and would make the assimilation exercise much easier: many of the accumulation
features are averaged out already at 100 m resolution. Assimilation of snow depth data at lower spatial resolution has been
found to give better results in previous synthetic ICESat-2 observations propagation with neural networks (Guidicelli et al.,
2023).

Simulating the snowpack with a physical-based model as we do with FSM2 provides inference on unobserved but socially
relevant variables (such as SWE) or fluxes (such as the snowmelt flux). The assimilation of fSCA has already been shown to
accurately reconstruct the peak SWE (Girotto et al., 2014), and here we show that adding snow depth observations from ICESat-
2 in the pool of assimilated snow data improves the peak basin-average snow depth. Despite not being able to demonstrate it
because of missing large-scale SWE validation measurements, we can postulate (based on results on SWE reconstruction
experiments using complete snow depth maps Margulis et al., 2019; Ma et al., 2023) that ICESat-2-retrieved snow depth can
improve the total water resources estimation for basins in complex terrain.

## 6 Conclusions and outlook

In this study, ICESat-2 snow depth observations along profiles were used for the first time to update high-resolution snowpack
simulations. We exploited the data assimilation system MuSA, which has capabilities to propagate information in space and
time developed in Alonso-González et al. (2023), to bridge the observations' sparsity. We perform a set of three experiments
where fSCA, snow depth, and both these observations were assimilated while keeping the assimilation algorithm and the prior
information the same, in order to evaluate the potential of the satellite laser altimeter ICESat-2 for updating seasonal snow
models.

We find that including two snow depth profiles in the set of assimilated observations improves the snowpack simulation in terms
of average snow depth, especially during the accumulation phase of the season – even though the snow depth profiles were
located completely outside the experimental catchment of Izas (55 ha), where we validate the experiments using drone-based
snow depth maps. Results show that the spatial patterns can only partially match the validation drone maps when assimilating
exclusively the snow depth profiles; as the snow depth pattern is very sensitive to the design of the spatially correlated prior



covariance which governs spatial propagation of information. Nevertheless, the joint assimilation of fSCA and snow depth from ICESat-2 bridged such limitations and performed best in terms of average snow depth as well as spatial distribution.

These findings indicate that the satellite ICESat-2 can be exploited to improve the state-of-the-art reanalysis generated by assimilating fSCA. Notably, the proposed workflow exploits globally-available datasets. Provided a high-resolution DEM –
which the geosciences community generally advocates the need for – ICESat-2's surface elevation measurements can be used to observe snow depth along profiles in inaccessible regions where snow amounts are still very hard to quantify. As multivariate DA techniques mature, the snow community can begin to exploit the plethora of snow observations (e.g. snow depth, fSCA, land surface temperature etc.) to constrain the snow models and to shed light on the unsolved snow hydrology problem of inferring the spatial and temporal distribution of SWE.

*Code and data availability.* The MuSA code ( version 2.1 used for the experiments can be found at Alonso-González et al., 2024). The ICESat-2 data was downloaded with SlideRule (Shean et al., 2023), the Sentinel-2 data with GEE (Gorelick et al., 2017). The complete input data for the experiments in the Izas basin can be found at (Mazzolini et al., 2024). Validation maps are available at (see Obs Alonso-González, 2022).

*Author contributions.* Conceptualization was by MM, with key contributions from DT and KA. Data curation: ICESat-2 data was by MM and
DT, Sentinel-2 data and ERA5 downscaling was by KA. Formal analysis was by MM and KA. Funding acquisition was by DT. Investigation was by MM and KA. Methodology was developed by EAG and KA, with key contributions from MM. Project administration was by DT. Software was by EAG, MM and KA. Supervision was by DT, KA, EAG and SW. Validation was by MM and KA. Visualization was by MM and DT. Writing – original draft preparation was lead by MM with key contributions by all co-authors.

*Competing interests.* The contact author has declared that none of the authors has any competing interests.

*Acknowledgements.* Marco Mazzolini and Désirée Treichler acknowledge funding from the Research Council of Norway (SNOWDEPTH project, contract 325519), DT additionally from the ESA project Glaciers_cci+ (4000127593/19/I-NB).
Kristoffer Aalstad acknowledges funding from the Research Council of Norway (Spot-On project, contract 301552), the ERC-2022-ADG under grant agreement No 01096057 GLACMASS, and an ESA CCI Research Fellowship (PATCHES project).
Esteban Alonso-González acknowledges funding from an ESA CCI Research Fellowship (SnowHotspots project)
We are grateful to NASA and USGS for the free provision of the ICESat-2 data, to Copernicus (and ECMWF) for the free provision of the Sentinel-2 (and ERA5) data. We are also grateful to all the developers of free software, mostly based on the Python programming language (Python Software Foundation, https://www.python.org/), used in this research such as: MuSA (Alonso-González et al., 2024) , xdem (Dehecq et al., 2021), SlideRule (Shean et al., 2023), xarray, etc..



Value-added data processed by CNES for the Theia data centre www.theia-land.frusing Copernicus products. The processing uses algorithms

developed by Theia's Scientific Expertise Centres.



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
