# Peer review of "Spatio-temporal snow data assimilation with the ICESat-2 laser altimeter"

_EGUsphere, 2024_

## Referee Comment (RC1)

Review of "Spatio-temporal snow data assimilation with the ICESat-2 laser altimeter"
by Mazzolini and others
submitted to The Cryosphere

**Summary**

The article presents the results of three data assimilation studies that incorporate (1) fractional snow-covered area from Sentinel-2, (2) snow depth from ICESat-2, and (3) both fractional snow-covered area and snow depth to determine which approach has the greatest improvement on modeled snow depths. The study is performed for a site in the Spanish Pyrenes where multiple drone-derived DEMs are available to assess the performance of the data-assimilated modeled outputs. The authors find that the inclusion of both fractional snow-covered area within the catchment and ICESat-2 snow depths from outside the catchment improve the model's ability to capture the distribution of snow depths in the catchment. Model performance is particularly improved during the snow accumulation season when ICESat-2 data are available, and degrades as the dominant processes that dictate snow distribution shift from accumulation to ablation processes.

The results of the study are interesting and the data assimilation approach appears to be a promising method to make the most use out of the sparse ICESat-2 tracks. I appreciate the detailed descriptions of agreement and disagreement between model outputs and observations. However, the writing can be a bit difficult to follow at times and I recommend that the authors make a number of revisions to the text and the figures in order to improve the manuscript.

**Major Comments**

1. There are several places where references are located early in sentences and it is unclear if they apply to the entire sentence, or where there is no reference provided but it should be. I've listed a few lines here but please make sure references are clear throughout the text:
   a. lines 17-18: Is Mott et al. (2018) for the entire sentence? If not, you need another reference to support everything that comes after its current location.
   b. Lines 105-107: You say that most DA research on snow has focused on temporal data assimilation with a few exceptions. You cite the exceptions but not the "most".
   c. Lines 118-119: The ATL03 data product is still be validated? You need to provide a reference here or remove the comment.
   d. Lines 129-133: You need some references here for information about the watershed, such as the fraction of precipitation that falls as snow and total precipitation.
   e. Lines 147-154: Neuenschwander et al. 2020 (https://doi.org/10.1016/j.rse.2020.112110) showed strong returns over snow for the weak beams. You should cite them here and I recommend you re-examine your weak beam data.

2. The introduction is very long. I understand that the authors feel like they need to provide background on a number of topics in order to justify and explain their work, but the reader is left wondering where they are going with the work because the introduction is so long. I recommend that the introduction is shortened considerably. You could base each paragraph around the following topics: (1) Why it is important to know snow depths across watersheds, (2) ICESat-2 looks like it can be used to estimate snow depths along its flight tracks, albeit with fairly large uncertainties, but we need a way to spatially and temporally extrapolate, (3) data assimilation techniques have shown promise for extrapolation, (4) this study explores data assimilation of ICESat-2 snow depths and Sentinel-2 snow-covered area. Then you can move a lot of the extra detail on techniques to measure snow depth (currently lines 33-50 and then 51-63 on ICESat-2 details) and various data assimilation techniques (currently lines 76-114) to the supplement so readers who are not familiar with those topics have a resource to lean on without bogging down the reader who knows plenty about those topics.

3. This seems like something pretty minor but all figures should have letter labels. Right now you need to refer to some of them by location and it would be a lot easier if they were all consistently lettered.

4. When describing the use of ensemble members in the data assimilation section, you state that you perturb some forcing variables. Why were those specific variables perturbed? You describe the shape of the perturbation but not the magnitude. What were the ranges of perturbation magnitudes and how were they selected?

5. The correlation length scale is stated as 1.5 in line 283. That is a unitless number. What does that equate to in terms of meters? Is it 30 m (1.5 grid cells)? Does that mean there is no correlation more than two times that distance away based on your explanation in line 280? There has been a lot of research on spatial correlation of snow depth and you need to tie your choice for this parameter to the literature. Right now you state that it was chosen to make the "size of the resulting neighborhoods acceptable". Acceptable to who or based on what? I recommend looking at https://agupubs.onlinelibrary.wiley.com/doi/10.1029/2020WR027343 and references therein regarding spatial correlation length scales. You could calculate variograms for your drone-based snow depths to determine the most appropriate scale for your study region.

6. The model has been described in more detail by the authors in their other publications that are cited in this manuscript, but it would be helpful to have a bit more detail in places. For example, in lines 286-287 it is stated that ICESat-2 snow depths are "spatially propagated" but fSCA information is not. What does this mean exactly? In the spatial propagation section you describe certain parameters that can be extracted from digital elevation models and how they are calculated over various distances. But there isn't a clear explanation of how the ICESat-2 data from outside the drone domain are spatially propagated. There is also no description of how the data are actually assimilated. Are the downscaled ERA5 data used to estimate snow patterns and then FSM2 adjusts tunable parameters to better match the fSCA maps? Is this what you are trying to explain in line 302? Everything is fairly disconnected as is and the reader needs to have a

general idea of how the modeling works without having to go back and read multiple other journal articles.

7. Figure 4-6: I really like that all the ensemble member's basin-averaged snow time series are shown in these figures, I like the color palette for the maps, and I like that the colors from the maps carry over into the histograms. That said, I think it is a bit of wasted space to keep showing the done map and histogram in every figure, especially since the map is also in Figure 1. I recommend showing a different map in Figure 1 to provide some added context and then merging these three figures into one multi-panel figure. In the merged figure, you could have the first column contain the legend for all the basin-averaged ensemble time series (which should be the same for all experiments but it is not) and then the drone peak snow map in the middle and drone peak snow histogram at the bottom. Then columns 2-4 would be the basin-averaged ensemble time series on top, snow depth map in the middle, and snow depth histogram on the bottom for experiments (C), (D), and (J). This would minimize redundancy and allow the reader to visually compare results a lot more easily.

**Minor Comments**
- Line 17: "by strong" and a comma after "processes"
- Line 18: "and metamorphism"
- Lines 23-24: Either remove this sentence with the Dozier reference or rephrase. Currently it doesn't fit with the rest of the paragraph.
- Lines 34-37: Something odd seems to have happened with the formatting here. The Foster reference seems to be thrown into the middle of this very long sentence and the sentence does not make sense.
- Lines 36-37: "where mountain regions are masked out" hangs on at the end of this sentence like an afterthought but it is an important point. Rephrase to emphasize.
- Line 49: The coarse footprint of ICESat?
- Line 52: Significantly better sensor characteristics than what?
- Line 56: Rephrase to "measurement error on flat terrain" instead of having part of the description in parentheses.
- Lines 64-83: There is a lot of extra information packed into parentheses in these paragraphs. Revise the sentences so that most information is written into the sentence. The use of parentheses makes it more difficult to read. For example, just say "obtaining statistically optimal estimates" on line 77.
- Line 89: Either keep "despite" or "thanks to" in the sentence but do not include them both with one in parentheses.
- Line 110: Rephrase to "In contrast, Alonso-González et al. (2023) have shown…"
- Line 113: What is "(see comment 6 ?)"?
- Line 114: Add a space after ICESat-2.
- Figure 1: The use of dashed lines to show the zoomed in areas is confusing because the ICESat-2 tracks are also dashed lines. I recommend using solid lines for the zooms or the tracks.

- Lines 141-160: There is a lot of information repeated here that was already in the introduction. You don't need to provide all the details of ICESat-2, just the ones that are important for your work. Then you only need to include them in one place.
- Line 170: Replace "harvests" with "hosts"
- Line 184: Why select 60%? How sensitive are your results to a different threshold?
- Line 195: Why bring up the orbit of the satellite here? What do you mean by "footprint"?
- Line 197-198: This is an incomplete thought. You filter out the cells with steep slopes?
- Figure 2: I like the idea of this figure but I cannot see the gray "ground photons" in the top panel. Consider revising the figure so the very top panel shows all the data, a middle panel shows all the photon differences with respect to the DEM, and the bottom panel stays as is. You would remove the right panel.
- Line 211: "20 m spatial resolution of the simulations 3.6"? I think the 3.6 should be totally removed but this also makes me realize that you describe all the data you will assimilate before you really describe the basic model. You might want to flip that order, moving 3.3 to the top of the methods, because you refer to the spatial resolution of the simulations before you describe them.
- Lines 213-217: These sentences on the uncertainty are very confusing. You list a sigma of 0.34 on line 213 and then again on line 217. Are these the same uncertainty metric or are they different metrics that miraculously have the same value? If they are the same, only list it once.
- Line 222: Replace "7" with "seven"
- Line 222: Do you mean that you select the most spatially detailed versions of parameterizations that you can use? Or the most mathematically complex? Or something else?
- Lines 223-226: For all of these parameterizations, I would simply say "as a function of" rather than "depending on", "influenced by", "diagnosed by", etc.
- Line 232: Why is 400 appropriate?
- Line 236: I am a bit confused by this sentence. If you are looking at figure 1, do you mean that you downscale for each cell in that large spatial domain in the left-most map? Based on your number of cells it seems unlikely. Your description does not sound like you only cover the small drone-based area. Do you also downscale to all the cells underlying all the ICESat-2 tracks or just those two highlighted tracks?
- Line 239: This is the first mention of "the prior". Presumably this means the model simulation with zero data assimilation. That needs to be defined either in this data assimilation section or in the more generic modelling section.
- Line 242: "log-normal" instead of "logit-normal"?
- Line 247: How is it "clearly non-linear"? Is that an interpretation based on manual inspection? Is that explained in previous literature?
- Figure 3: Rearrange the panels, and add letter labels, so that they go from left to right according to the order that each variable is described in the text: TPI, Sx, then CSMD.
- Line 311: Explicitly state their measurement uncertainty.

- Line 321: "selected by the median operator"? Does this mean the map with the median snow depth out of all ensemble members?
- Lines 409-242: Here is where "the prior" comes up repeatedly but it was never well defined. Please address my earlier comment so the reader can more easily follow this discussion.
- Lines 475-490: I am not a data assimilation expert so found the repeated use of "hyper" to be confusing here. You mention hyperparameters as earlier in the text, which is fine, but you also refer to a hyperprior and say the experiments are almost hyper. Does this all just mean high spatial resolution?

---

## Author Comment (AC1)

**REVIEWER #1**

**Major Comments:**

COMMENT # 1.1

*The article presents the results of three data assimilation studies that incorporate (1) fractional snow-covered area from Sentinel-2, (2) snow depth from ICESat-2, and (3) both fractional snow-covered area and snow depth to determine which approach has the greatest improvement on modeled snow depths. The study is performed for a site in the Spanish Pyrenes where multiple drone-derived DEMs are available to assess the performance of the data-assimilated modeled outputs. The authors find that the inclusion of both fractional snow-covered area within the catchment and ICESat-2 snow depths from outside the catchment improve the model's ability to capture the distribution of snow depths in the catchment. Model performance is particularly improved during the snow accumulation season when ICESat-2 data are available, and degrades as the dominant processes that dictate snow distribution shift from accumulation to ablation processes. The results of the study are interesting and the data assimilation approach appears to be a promising method to make the most use out of the sparse ICESat-2 tracks. I appreciate the detailed descriptions of agreement and disagreement between model outputs and observations. However, the writing can be a bit difficult to follow at times and I recommend that the authors make a number of revisions to the text and the figures in order to improve the manuscript.*

**Reply:**

We appreciate the Reviewer's interest, time, and insightful comments. We are very grateful for the constructive suggestions that have helped us to make study better and more readable. The following provides a point-by-point response to the Reviewer's comments.

    **Major comments:**

COMMENT # 1.2

*There are several places where references are located early in sentences and it is unclear if they apply to the entire sentence, or where there is no reference provided but it should be. I've listed a few lines here but please make sure references are clear throughout the text.*

    *a. lines 17-18: Is Mott et al. (2018) for the entire sentence? If not, you need another reference to support everything that comes after its current location.*

    *b. Lines 105-107: You say that most DA research on snow has focused on temporal data*

*assimilation with a few exceptions. You cite the exceptions but not the "most".*

c. *Lines 118-119: The ATL03 data product is still be validated? You need to provide a reference here or remove the comment.*

d. *Lines 129-133: You need some references here for information about the watershed, such as the fraction of precipitation that falls as snow and total precipitation.*

e. *Lines 147-154: Neuenschwander et al. 2020 (https://doi.org/10.1016/j.rse.2020.112110) showed strong returns over snow for the weak beams. You should cite them here and I recommend you re- examine your weak beam data*

**Reply:**

We thank the Reviewer for the detailed reading and checked the whole text for inappropriate or unclear references located early in sentences. Following are the detailed replies to the specific phrases:

a. The reference is for the entire sentence, so we moved it to the end.

b. Added a relevant reference at the end of the phrase where it is stated that most snow DA research focuses on purely temporal DA (Girotto et al., 2020).

c. We find that ATL03 data can provide snow depths with ca. decimetric accuracy (see the conference presentation: Treichler et al., 2023). Some of the authors also have a publication in preparation about the validation of ICESat-2 measurements with drone data. We will either refer to the publication if it becomes available in time, or remove this part.

d. Moved the citation to Revuelto et al. (2017) to the end of the sentences that refer to that paper.

e. This is a valid point, we agree with the Reviewer that the weak beams may contain valuable information over bright, snow-covered terrain and should thus not be generally discarded. In Neuenschwander et al. (2020), ATL08 data is validated in the Finnish boreal forests, a terrain with little topographic relief. ATL08 splits photons in 100 m, while we work with 20 m cells, diminishing by a factor of five the number of photons available in one cell. There are also strong terrain differences between the mentioned study and the Izas study area, where a large average slope can negatively impact snow depth estimation in case of horizontal geolocation inaccuracy (3 to 4 meters; Magruder et al., 2021). Finally, only a fourth of the photons are available for the weak beams profiles (strong to weak beam energy ratio is 4; Neumann et al., 2019) compared to strong beams,

making the statistical estimation of snow depth less reliable. Such considerations made us discard the weak beams observations, but we consider adding them in the revised manuscript if they prove useful to the goals of the study.

COMMENT # 1.3

*The introduction is very long. I understand that the authors feel like they need to provide background on a number of topics in order to justify and explain their work, but the reader is left wondering where they are going with the work because the introduction is so long. I recommend that the introduction is shortened considerably. You could base each paragraph around the following topics: (1) Why it is important to know snow depths across watersheds, (2) ICESat-2 looks like it can be used to estimate snow depths along its flight tracks, albeit with fairly large uncertainties, but we need a way to spatially and temporally extrapolate, (3) data assimilation techniques have shown promise for extrapolation, (4) this study explores data assimilation of ICESat-2 snow depths and Sentinel-2 snow-covered area. Then you can move a lot of the extra detail on techniques to measure snow depth (currently lines 33-50 and then 51-63 on ICESat-2 details) and various data assimilation techniques (currently lines 76-114) to the supplement so readers who are not familiar with those topics have a resource to lean on without bogging down the reader who knows plenty about those topics.*

**Reply:**

We made the introduction shorter, and we thank the Reviewer for suggesting a nice paragraph structure for this section. We believe the introduction is now easier to read thanks in part to this suggestion.

COMMENT # 1.4

*This seems like something pretty minor but all figures should have letter labels. Right now you need to refer to some of them by location and it would be a lot easier if they were all consistently lettered.*

**Reply:**

We thank the Reviewer for this practical comment. We added the labels to the panels, and improved the references to the figures in the text.

COMMENT # 1.5

*When describing the use of ensemble members in the data assimilation section, you state that you perturb some forcing variables. Why were those specific variables perturbed? You describe the shape of the perturbation but not the magnitude. What were the ranges of perturbation magnitudes and how were they selected?*

**Reply:**

We chose these variables (temperature, precipitation and longwave radiation) because perturbing them allows us to account for uncertainty in the forcing data and obtain an ensemble with a variety of possible scenarios for the seasonal snow evolution:

- the precipitation perturbation allows to locally adjust the snow accumulation or removal processes that are not explicitly modelled (e.g. wind redistribution);

- the perturbation to the temperature allows for different precipitation phase scenarios (liquid or solid) and modulates the sensible heat flux;

- the perturbation to the longwave radiation modulates the radiative part of the energy balance, thus modifying the internal energy and melt processes in the snowpack.

One should note that it is an equally acceptable choice to abandon the forcing formulation of DA and to instead infer a set of internal parameters of the snow model (FSM2). However, given our purpose of experimenting with a new set of remotely sensed observations, the choice of a reasonable set of parameters or another is irrelevant for the scope of the paper. The parameters for the perturbation applied to the forcing to generate the prior were previously only available from the configuration files uploaded in the zenodo repository Mazzolini et al. (2024), but to clarify we have now added the following Table 1 to Section 3.4 to fully describe the prior perturbation parameter distribution employed for each of the variables. The parameters were chosen based on previous studies' values (Alonso-González et al., 2022, 2023).

**Changes:**

In the presented experiments, the perturbed forcing variables are air temperature, precipitation and downwelling longwave radiation. The perturbation parameters are time-invariant throughout the water year, and  are extracted from a logit-normal distribution  whose prior hyperparameters can be seen in Table 1. We choose this distribution over a log-normal or a Gaussian distribution as the logit-normal restricts the perturbation within defined ,

(Guidicelli et al., 2023)upper and lower bounds, in contrast the other distributions which would have respectively only one or no bounds (Aitchison and Shen, 1980) . The nature of the perturbation is multiplicative for the precipitation (in part to prevent non-physical negative values) and additive for the other variables.

Table 1: Hyperparameters for extraction of the logit-normally distributed prior perturbation parameters. A note for the DA-expert reader is that the hyperprameters $\mu$ and $\sigma$ are the mean and standard deviation, respectively, of the associated Gaussian distributions that the logit-transformed prior perturbation parameters follow. Numerical entries without units are implicitly dimensionless.

| Perturbed Variable | Type | $\mu$ | $\sigma$ | Lower bound | Upper bound |
|---|---|---|---|---|---|
| Precipitation | Multiplicative | $-0.9$ | $0.7$ | $0.1$ | $5$ |
| Temperature | Additive | $0$ | $0.5$ | $-8\,\mathrm{K}$ | $8\,\mathrm{K}$ |
| Longwave radiation | Additive | $0$ | $0.5$ | $-8\,\mathrm{Wm}^{-2}$ | $8\,\mathrm{Wm}^{-2}$ |

COMMENT # 1.6

*The correlation length scale is stated as 1.5 in line 283. That is a unitless number. What does that equate to in terms of meters? Is it 30 m (1.5 grid cells)? Does that mean there is no correlation more than two times that distance away based on your explanation in line 280? There has been a lot of research on spatial correlation of snow depth and you need to tie your choice for this parameter to the literature. Right now you state that it was chosen to make the "size of the resulting neighborhoods acceptable". Acceptable to who or based on what? I recommend looking at https://agupubs.onlinelibrary.wiley.com/doi/10.1029/2020WR027343 and references therein regarding spatial correlation length scales. You could calculate variograms for your drone-based snow depths to determine the most appropriate scale for your study region.*

**Reply:**

We thank the Reviewer for this astute question. This Comment, together with the following Comment and Reviewer 2 Comment 1.5, made it clear that we need to expand the explanation of the spatial transfer of information from the available ICESat-2 observations. We offer here an answer that will be incorporated in Section 3.5 of the revised paper.

The correlation length scale is unitless because it does not correspond to a geo-

[Figure]

Figure 1: *Panel a): scatterplot depicting the position of the cells from the drone maps in the feature space. This space – created with TPI and CSMD – is adopted to define the similarity between cells. The points are colored according to the snow depth observed with the drone. Panel b): ICESat-2 snow depth observations in the extended catchment, displayed in feature space, with snow depth-based coloring. The cross represents one cell from the drone domain where a snow depth of 150 cm was measured. The solid points are ICESat-2 data points included in the neighbourhood for this cell, with their size proportional to the correlation ρ.*

graphical distance but to a distance in standardized multi-dimensional feature space, i.e. the similarity with regard to TPI and CSMD. In simpler words, information is transferred between cells that have similar convexity and average snow disappearence date.

To guide the reader in understanding this concept, we add to the revised manuscript an example showing how the neighbourhood is defined in the feature space, with the help of Figure 1 that you find below. Focus on panel b): there we exemplify a situation where a cell in the catchment with drone data – depicted therein with a cross – has to be updated. The solid points in the scatterplot are selected to be part of the neighbourhood, and all of them have influence on the Kalman update (see step 11 in Algorithm 1 of Alonso-González et al., 2023), used to update the local ensemble of the target grid cell. As cells closer in feature space to the target cell should have a larger influence, their ρ is larger, which can be appreciated by looking at the size of the scatter points.

**Comment # 1.7**

*The model has been described in more detail by the authors in their other publications that are cited in this manuscript, but it would be helpful to have a bit more detail in- places. For ex-*

*ample, in lines 286-287 it is stated that ICESat-2 snow depths are "spatially propagated" but fSCA information is not. What does this mean exactly? In the spatial propagation section you describe certain parameters that can be extracted from digital elevation models and how they are calculated over various distances. But there isn't a clear explanation of how the ICESat-2 data from outside the drone domain are spatially propagated. There is also no description of how the data are actually assimilated. Are the downscaled ERA5 data used to estimate snow patterns and then FSM2 adjuststunable parameters to better match the fSCA maps? Is this what you are trying to explain in line 302? Everything is fairly disconnected as is and the reader needs to have a general idea of how the modeling works without having to go back and read multiple other journal articles.*

**Reply:**

We thank the Reviewer for this constructive critique. The detailed explanation of the unclear topics greatly helped us to expand Section 3.5. Below, we try to answer the Reviewer's individual questions:

- *"For example, in lines 286-287 it is stated that ICESat-2 snow depths are "spatially propagated" but fSCA information is not. What does this mean exactly?"* In purely temporal (no spatial propagation) ensemble-based data assimilation, the observation located in a cell is used for a comparison with the corresponding observable state variable to compute a direct update to the prior's forcing perturbation parameter via an (ensemble) Kalman analysis step. For a more detailed explanation of this, see equation 6.37 to 6.39 in Evensen et al. (2022). In spatio-temporal data assimilation, the observations taken into account for updating the state in a certain cell, are all those that are located in the neighbourhood of the cell (see the updated Section 3.5 for an explanation of how the neighbourhood is defined). However, fSCA observations are spatially complete: hence every cell (or almost) has multiple observations of fSCA. As a consequence, we do not use other fCSA observations other than the local ones located in the cell itself to update the parameters in that cell. We hope to have clarified this in the updated version of Section 3.5

- *"But there isn't a clear explanation of how the ICESat-2 data from outside the drone domain are spatially propagated."* In contrast to what we just said about fCSA, we stated that we spatially propagate information from ICESat-2 observations. This means that when we update the perturbation's parameter for any cell in the experimental catchment, part of the ICESat-2 observations (the ones falling into its neighbourhood) are used. Algorithm 1 in Alonso-González et al. (2023) contains the practical equations used for the spatio-temporal assimilation with the DES-MDA scheme. Here we highlight that when a cell is updated, the observations located in a cell which is close in the feature space will have a large correlation ρ, and hence have a large influence on the update, while observations located in a cell which is far (but still inside the neighbourhood) will have a small ρ and hence have limited influence in the update. We hope to have clarified this in the updated version of Section 3.5, and a visual example of this is offered in Figure 1.

- *"There is also no description of how the data are actually assimilated."* We use the assimilation algorithm DES-MDA, which is an iterative ('multiple data assimilation' or MDA) and so-called deterministic (non stochastic) version of the ES. In short, this iterative assimilation algorithm performs a form of likelihood tempering by inflating the observation's standard deviation, so as to divide the update in multiple iterative smaller update steps – without violating Bayes' theorem – and leading to better performance with nonlinear models. The deterministic (also known as square root) nature of this ensemble Kalman scheme simplifies the numerics since it does not involve perturbations in observation space while also leading to improved performance with a smaller ensemble size. We will modify Section 3.4 to point the reader to Alonso-González et al. (2022) and Alonso-González et al. (2023) where the practical implementation of this algorithm is described.

- *"Are the downscaled ERA5 data used to estimate snow patterns and then FSM2 adjusts the tunable parameters to better match the fSCA maps? Is this what you are trying to explain in line 302?"* ERA5 data is topographically downscaled and used (together with the prior perturbation parameters) to run an ensemble of FSM2 simulations in every cell. In the assimilation of fSCA it is possible to imagine every cell independently from its neighbours. As we are using what is usually called forcing formulation of DA, the updates (or you can think of them as adjustments) are applied directly to the forcing perturbation's parameters and subsequently indirectly to the model states by re-running the model with updated perturbed forcing. We try to give a practical example with how the assimilation of fSCA would influence the perturbation parameters. First, an ensemble of FSM2 simulations is run over a cell. We will take the time point of the observation, and use the distribution (represented by the ensemble) of predicted fSCA to compare with the observed fCSA. At this point an update of the perturbation parameters is computed in such a way that the updated perturbations will be sampled around those perturbations values that led to a predicted fSCA similar to the observed fSCA. The ensemble simulation with the updated perturbation parameters will lead to a fSCA simulation closer to the observation. While this is only an example, DA algorithms such as the ES-MDA we use

have already been tested for assimilating various snow observations in many studies (see Alonso-González et al., 2022, and references therein).

COMMENT # 1.8

*Figure 4-6: I really like that all the ensemble member's basin-averaged snow timeseries are shown in these figures, I like the color palette for the maps, and I like that the colors from the maps carry over into the histograms. That said, I think it is a bit of wasted space to keep showing the done map and histogram in every figure, especially since the map is also in Figure 1. I recommend showing a different map in Figure 1 to provide some added context and then merging these three figures into one multi-panel figure. In the merged figure, you could have the first column contain the legend for all the basin-averaged ensemble time series (which should be the same for all experiments but it is not) and then the drone peak snow map in the middle and drone peak snow histogram at the bottom. Then columns 2-4 would be the basin-averaged ensemble time series on top, snow depth map in the middle, and snow depth histogram on the bottom for experiments (C), (D), and (J). This would minimize redundancy and allow the reader to visually compare results a lot more easily.*

**Reply:**

We thank the Reviewer for this Comment and agree that the multi-panel Figure 2 (included below) is a better way to compare the three experiments' results while avoiding redundancy. The map in the original manuscript's Figure 1 has a finer resolution than the one shown in the results, we believe it provides the most useful information in this context compared to snow depth maps from other time steps.

**Minor Comments:**

COMMENT # 1.9

*Line 17: "by strong" and a comma after "processes"*

**Reply:**

Changed, see answer for Comment 1.10

COMMENT # 1.10

*Line 18: "and metamorphism"*

**Reply:**

[Figure]

Figure 2: *Panels a), b) and c) show the prior (gray) and posterior ensemble simulations (colored) as catchment-average snow depths over the whole water year for experiment **C**, **D**, **J**, respectively. The black points are the drone-based snow map averages serving as validation data. The blue and orange stars show the timing of the fSCA and/or snow depth observations, respectively, that are assimilated in each experiment. Panels d), e) and f): simulated snow depths are shown as heat maps for the 11.03.2020 (date shown with a vertical dashed line in panels above) for a representative ensemble member that is nearest to the ensemble median catchment average snow depth for experiment **C**, **D**, **J**, respectively. Panel d): the corresponding 11.03.2020 drone-based snow depth heat map at the model's spatial resolution (20 m). Panels h), j), k) and l) below showing snow depth histograms corresponding to the (and defining the color maps of) the heat maps in the panels above.*

The introduction has largely changed and the corresponding part is rewritten as follows:

**Changes:**

Seasonal snow is  a crucial variable for sustaining human life and an essential climate regulator (Sturm et al., 2017). It is characterized by strong spatial and temporal variability  which arises from several processes such as preferential deposition, wind transport, differential radiation and heat fluxes  and metamorphism (Mott et al., 2018).

COMMENT # 1.11

*Lines 23-24: Either remove this sentence with the Dozier reference or rephrase. Currently it doesn't fit with the rest of the paragraph.*

**Reply:**

Removed.

COMMENT # 1.12

*Lines 34-37: Something odd seems to have happened with the formatting here. The Foster reference seems to be thrown into the middle of this very long sentence and the sentence does not make sense.*

**Reply:**

We thank the Reviewer for spotting this. This part of the introduction has been greatly condensed.

COMMENT # 1.13

*Lines 36-37: "where mountain regions are masked out" hangs on at the end of this sentence like an afterthought but it is an important point. Rephrase to emphasize.*

**Reply:**

The description of the state-of-the-art snow remote sensing has been removed from the introduction as suggested in Comment 1.3.

COMMENT # 1.14

*Line 49: The coarse footprint of ICESat?*

**Reply:**

Yes, that's what we meant. This part of the introduction has been rewritten.

COMMENT # 1.15

*Line 52: Significantly better sensor characteristics than what?*

**Reply:**

We meant that ICESat-2 has better characteristics than its predecessor ICESat in terms of spatial resolution. However, we removed this phrase from the introduction.

COMMENT # 1.16

*Line 56: Rephrase to "measurement error on flat terrain" instead of having part of the description in parentheses.*

**Reply:**

Thank you, we modified this as follows:

**Changes:**

The geolocated photons have a centimetric vertical measurement error  on flat terrain (Markus et al., 2017)  , while the horizontal accuracy is estimated at 3 to 4 m (Magruder et al., 2021).

COMMENT # 1.17

*Lines 64-83: There is a lot of extra information packed into parentheses in these paragraphs. Revise the sentences so that most information is written into the sentence. The use of parentheses makes it more difficult to read. For example, just say "obtaining statistically optimal estimates" on line 77.*

**Reply:**

We improved readability in the revised manuscript by avoiding parentheses: We removed the phrase between parentheses at line 65-66 and removed the parentheses at line 77. Lines 80-84 were removed in the larger restructuring of the introduction.

COMMENT # 1.18

*Line 89: Either keep "despite" or "thanks to" in the sentence but do not include them both with one in parentheses*

**Reply:**

This phrase was removed in the larger restructuring of the introduction.

COMMENT # 1.19

*Line 110: Rephrase to "In contrast, Alonso-González et al. (2023) have shown..."*

**Reply:**

Rephrased as suggested.

COMMENT # 1.20

*Line 114: Add a space after ICESat-2.*

**Reply:**

We thank the Reviewer for spotting it, added.

COMMENT # 1.21

*Figure 1: The use of dashed lines to show the zoomed in areas is confusing because the ICESat-2 tracks are also dashed lines. I recommend using solid lines for the zooms or the tracks.*

**Reply:**

We thank the Reviewer for this suggestion, we will change the line style in the figure to improve readability.

COMMENT # 1.22

*Lines 141-160: There is a lot of information repeated here that was already in the introduction. You don't need to provide all the details of ICESat-2, just the ones that are important for your work. Then you only need to include them in one place.*

**Reply:**

The Reviewer is right, in the revised manuscript we take care to avoid repetitions. We introduce ICESat-2 briefly in the introduction and provide further details essential to understand the method in the data section.

COMMENT # 1.23

*Line 170: Replace "harvests" with "hosts"*

**Reply:**

Replaced.

COMMENT # 1.24

*Line 184: Why select 60%? How sensitive are your results to a different threshold?*

**Reply:**

The 60% mark was chosen after inspecting the photon snow depth profiles visually and applying several thresholds. For the two profiles in our study, it was found to effectively remove noise photons having few neighbours while preserving most photons reflected from the snow surface – as photons cluster along the continuous surface. Noise photons stemming from the atmosphere, double bounces, and photons scattered in the snowpack before being reflected back to ICESat-2 are removed. Keeping a larger proportion of the photons leads to a more complete profile, but at the cost of a increasing the number of outliers, and keeping fewer photons would produce data gaps in the snow surface profile. The profiles used in this study are relatively short and were thus thoroughly checked for quality and completeness. However, we recommend a sensitivity test or possibly multiple thresholds for future applications in larger study sites where ICESat-2 profiles from different overpasses are included and manual checks of all data is not possible. The 60% was as an appropriate proportion for the specific average slope and solar radiation at the time of acquisition but may not fit other sites and acquisition conditions (e.g., night acquisitions, haze or blowing snow conditions, different slopes or snow reflectivity).

COMMENT # 1.25

*Line 195: Why bring up the orbit of the satellite here? What do you mean by "footprint"?*

**Reply:**

The orbit was mentioned to remind the reader that the profiles are not directed north-south but slightly inclined. The footprint is the size of the Earth's surface illuminated by one pulse of the ICESat-2 laser (ca. 14 m in diameter), we adjusted the text in section 2 to clarify this. Here, we point to this to remind the reader of the width of the profile. Line 195 has been rewritten to clarify how the photons, grid and profile

geometries fit together.

**COMMENT # 1.26**

*Line 197-198: This is an incomplete thought. You filter out the cells with steep slopes?*

**Reply:**

Apologies, this phrase was not complete.

**Changes:**

In addition, also the cells with an average slope larger than 40° are filtered out, as the horizontal positioning uncertainty makes snow depth retrievals less reliable for steep terrain.

**COMMENT # 1.27**

*Figure 2: I like the idea of this figure but I cannot see the gray "ground photons" in the top panel. Consider revising the figure so the very top panel shows all the data, a middle panel shows all the photon differences with respect to the DEM, and the bottom panel stays as is. You would remove the right panel.*

**Reply:**

Thank you for this suggestion. We will revise the figure.

**COMMENT # 1.28**

*Line 211: "20 m spatial resolution of the simulations 3.6"? I think the 3.6 should be totally removed but this also makes me realize that you describe all the data you will assimilate before you really describe the basic model. You might want to flip that order, moving 3.3 to the top of the methods, because you refer to the spatial resolution of the simulations before you describe them.*

**Reply:**

We thank the reviewer, we removed the 3.6 reference from line 211 and also flip the subsections, as we agree it makes the paper more readable.

**COMMENT # 1.29**

> *Lines 213-217: These sentences on the uncertainty are very confusing. You list a sigma of 0.34 on line 213 and then again on line 217. Are these the same uncertainty metric or are they different metrics that miraculously have the same value? If they are the same, only list it once.*

**Reply:**

Yes, the metrics were the same. We aimed at showing how we computed our uncertainty estimate. We remove 0.34 from line 217 and simplify this part:

**Changes:**

We estimate the observation error for the fSCA retrievals at 20 m resolution to be $\sigma = 0.34$.  Independent validation estimated the observation error $\sigma_N$ at 100 m resolution to be equal to $\sigma_N = 0.07$ (see Table 2 Aalstad et al., 2020). We obtained our 20 m $\sigma$ estimate using that the error at coarser  resolutions should increase at higher resolutions according to the central limit theorem  through $\sigma = \sigma_N \sqrt{N}$, where $N = 25$ is the number of independent 20 m cells  contained in the coarser (100 m) validation.

COMMENT # 1.30

> *Line 222: Replace "7" with "seven"*

**Reply:**

Replaced.

COMMENT # 1.31

> *Line 222: Do you mean that you select the most spatially detailed versions of parameterizations that you can use? Or the most mathematically complex? Or something else?*

**Reply:**

FSM2 has two or three levels of representation for each of the physical process we mention in the lines 223-226, where one is very simple and the other is more complex. To explain this better we change the phrase to:

**Changes:**

 In FSM2, seven physical processes are  represented with multiple available process parameterizations. We choose the most complex representation for all the processes to obtain a more comprehensive snowpack ensemble simulation.

COMMENT # 1.32

*Lines 223-226: For all of these parameterizations, I would simply say "as a function of" rather than "depending on", "influenced by", "diagnosed by", etc.*

**Reply:**

Agreed, we change the phrasing except for the statement about the turbulent fluxes where the inputs to the coupled functional relationships in the Monin-Obukhov similarity theory are omitted for brevity.

**Changes:**

These parametrizations are: albedo decay with elapsed time since the last significant snowfall, thermal conductivity  as a function of snow density, density  as a function of overburden and metamorphism, turbulent fluxes diagnosed using the Monin-Obukhov similarity theory, and melt-water percolation  as a function of gravitational drainage, fractional snow cover asymptotic to snow depth.

COMMENT # 1.33

*Line 232: Why is 400 appropriate?*

**Reply:**

We thank the Reviewer for the question. We downscale the ERA5 forcing to the extended catchment (see panel a) of Figure 1 in the original manuscript), approximately sized 5 km by 3 km. The topographical downscaling works by grouping the cells in a number of clusters with similar topographic metrics such as complexity, aspect, slope and others. You can see more details about this in Fiddes and Gruber (2012). In that work, it was shown that an entire ERA5 grid cell (25 km) was split in as little as 100 clusters while capturing first order hillslope scale variability in the atmospheric forcing and the resulting cryospheric simulations. Our choice of 400 clusters is a considerably larger number and is chosen to confidently ensure that hillslope-induced variability in the forcing data is captured for a considerably smaller area than an

entire ERA5 grid cell. Note that this number doesn't influence the later snow simulations, but is only used for downscaling to generate the atmospheric forcing.

C OMMENT # 1.34

*Line 236: I am a bit confused by this sentence. If you are looking at figure 1, do you mean that you downscale for each cell in that large spatial domain in the left-most map? Based on your number of cells it seems unlikely. Your description does not sound like you only cover the small drone-based area. Do you also downscale to all the cells underlying all the ICESat-2 tracks or just those two highlighted tracks?*

**Reply:**

We thank the Reviewer for the question. TopoSCALE downscales the forcing in a semi-distributed manner, so the forcing is downscaled for a number of clusters with a certain topographic signature each (400 here), representing the entire domain. These can then be mapped back to the full grid — but only some cells are simulated by the snow model, the drone-mapped area and the two ICESat-2 highlighted tracks.

**Changes:**

The obtained semi-distributed forcing is then mapped back to  a 20 m fully distributed grid  covering the whole extended domain providing forcing data for a selection of cells. We now select only the cells covering the drone maps in the Izas experimental catchment (solid black line in Figure 1) and the grid cells in the extended domain that are intersected by the ICESat-2 tracks (blue lines in Figure 1), summing up to a total of ~ 1900 cells for which the FSM2 model is run. ...

C OMMENT # 1.35

*Line 239: This is the first mention of "the prior". Presumably this means the model simulation with zero data assimilation. That needs to be defined either in this data assimilation section or in the more generic modelling section.*

**Reply:**

We thank the Reviewer for noticing this. We added a definition at the beginning of Section 3.4 that can be seen in the answer to Comment 1.41.

*Line 242: "log-normal" instead of "logit-normal"?*

**Reply:**

We have used a logit-normal (also known as a logistic normal) distribution Aitchison and Shen (1980), which is a distribution with both upper and lower bounds, while a log-normal distribution would only have a lower bound — which is not as useful for parameters that have multiple physical constraints. A table showing the distribution hyper-parameters was added to the manuscript and in the reply to Comment 1.5.

**Changes:**

...The perturbation parameters are time-invariant throughout the water year, and the prior perturbation parameters are extracted via transformations from a logit-normal distribution rather than Gaussian , to restrict whose prior hyper-parameters can be seen in Table 1. We choose this distribution over a log-normal or a Gaussian distribution as the logit-normal restricts the perturbation within defined bounds (Aalstad et al., 2018; Guidicelli et al., 2023)upper and lower bounds, in contrast the others which would have respectively only one or no bounds (Aitchison and Shen, 1980).

Comment # 1.37

*Line 247: How is it "clearly non-linear"? Is that an interpretation based on manual inspection? Is that explained in previous literature?*

**Reply:**

The modeled relation between input forcing and observable state (fSCA or snow depth) realized through FSM2 is non-linear in several ways. One example is the snow/water phase change at zero degrees, as can be illustrated by the following example of the relation between longwave radiation and snow depth. A positive perturbation (i.e. more incoming radiation) has no effect on the snow depth until the point in the season when the snowpack becomes isothermal and starts melting, while it has a large impact after this point in time. Another example is the relationship between temperature and precipitation and their non-linear effect on snow depth: a positive temperature perturbation leading to liquid precipitation will cause snowdepth to remain null or decrease due to changes in density and crystal structure, while for lower temperatures (and, consequently, solid precipitation) a perturbation will cause no further increase/decrease in snowdepth. Another way to verify the

non-linearity can be found in Essery (2015), where one can clearly see how many non-linear functions are used to define the relation between atmospheric input and observable states such as fSCA and snow depth.

*Figure 3: Rearrange the panels, and add letter labels, so that they go from left to right according to the order that each variable is described in the text: TPI, Sx, then CSMD.*

**Reply:**

Changed

*Line 311: Explicitly state their measurement uncertainty.*

**Reply:**

We can estimate the uncertainty for the drone maps at 20 m resolution with the central limit theory, obtaining conservatively a centimetric accuracy. Note that this estimation implies the hypothesis of independent identically distributed measurement errors, which is not respected when the errors are spatially correlated. So we degraded the estimation of 1 order of magnitude, obtaining a "conservative" estimate.

**Changes:**

Their measurement error  _can conservatively be estimated at about 1 cm as we have resampled_ the snow depth maps to the modelling resolution (from 1 m to 20 m) with the averaging operator.

*Line 321: "selected by the median operator"? Does this mean the map with the median snow depth out of all ensemble members?*

**Reply:**

Yes, the Reviewer is correct. We realize the phrasing here was a bit complicated, and tried to clarify this in the text. The spatially distributed median of all ensemble members is not a representative model output since it would show values from a different member for each point in space, hence mixing up different model runs. To pick a

single model ensemble member, we compute the spatial average of all the member's maps for the 11th of March, and then pick the median snow depth member. This ensures that the reference member is representative for the model ensemble mean while corresponding to an actual model run.

**Changes:**

Since the result of the DA problem is a spatially correlated ensemble representing a statistical distribution, we show one single ensemble member simulation in order to appreciate the spatial structure embedded in the simulation.  To choose the representative member, we first select the simulation state on 11th of March as this is the closest drone acquisition to the peak-SWE. Then we spatially average the ensemble members and pick the  median of those spatial averages.

COMMENT # 1.41

*Lines 409-242: Here is where "the prior" comes up repeatedly but it was never well defined. Please address my earlier comment so the reader can more easily follow this discussion.*

**Reply:**

We added a definition of prior simulation at the beginning of Section 3.4 so the reader can now follow:

**Changes:**

... Therein, the prior  distribution – a probabilistic distribution representing uncertainty over the system's state and parameter space before observations are taken into account – is represented by the spread of  a finite collection of samples known as ensemble members.  This spread in terms of basin-average snow depth can be seen in the gray trajectories of panels a), b) and c) of Figure 2. Each prior ensemble member is an FSM2 simulation obtained by perturbing a selection of forcing variables.

COMMENT # 1.42

*Lines 475-490: I am not a data assimilation expert so found the repeated use of "hyper" to be confusing here. You mention hyperparameters as earlier in the text, which is fine, but you also refer to a hyperprior and say the experiments are almost hyper. Does this all just mean high spatial resolution?*

**Reply:**

We thank the Reviewer for this comment, hyper was used with two different meanings for model parameters and spatial resolution. In modelling, hyperparameters refer to high level parameters controlling the statistical distribution of lower level parameters. In the paper, this is used for the prior hyperparameters that control the extraction of the spatially correlated prior. Due to computational limitations snow models are often run at a spatial resolution that may seem coarse to the observation/field measurements community. In this context, 20 m corresponds to a very high resolution that is often referred to as hyper-resolution. We opt for keeping these two meanings, but the reader will be able to distinguish the two contexts as in the first is written without hyphenation (hyperparameter), while we add the hyphen when talking about spatial resolution (e.g. hyper-resolution).

**References**

Aalstad, K., Westermann, S., Schuler, T., Boike, J., and Bertino, L.: Ensemble-based assimilation of fractional snow-covered area satellite retrievals to estimate the snow distribution at Arctic sites, The Cryosphere, 12, 247–270, https://doi.org/10.5194/tc-12-247-2018, 2018.

Aalstad, K., Westermann, S., and Bertino, L.: Evaluating satellite retrieved fractional snow-covered area at a high-Arctic site using terrestrial photography, Remote Sensing of Environment, 239, 111 618, https://doi.org/10.1016/j.rse.2019.111618, 2020.

Aitchison, J. and Shen, S. M.: Logistic-Normal Distributions: Some Properties and Uses, Biometrika, 67, 261–272, https://doi.org/10.2307/2335470, publisher: [Oxford University Press, Biometrika Trust], 1980.

Alonso-González, E., Aalstad, K., Baba, M. W., Revuelto, J., López-Moreno, J. I., Fiddes, J., Essery, R., and Gascoin, S.: The Multiple Snow Data Assimilation System (MuSA v1.0), Geoscientific Model Development, 15, 9127–9155, https://doi.org/10.5194/gmd-15-9127-2022, 2022.

Alonso-González, E., Aalstad, K., Pirk, N., Mazzolini, M., Treichler, D., Leclercq, P., Westermann, S., López-Moreno, J. I., and Gascoin, S.: Spatio-temporal information propagation using sparse observations in hyper-resolution ensemble-based snow data assimilation, Hydrology and Earth System Sciences, 27, 4637–4659, https://doi.org/10.5194/hess-27-4637-2023, 2023.

Essery, R.: A factorial snowpack model (FSM 1.0), Geoscientific Model Development, 8, 3867–3876, https://doi.org/10.5194/gmd-8-3867-2015, 2015.

Evensen, G., Vossepoel, F. C., and van Leeuwen, P. J.: Data Assimilation Fundamentals: A Unified Formulation of the State and Parameter Estimation Problem, Springer Textbooks in Earth Sciences, Geography and Environment, Springer International Publishing, Cham, https://doi.org/10.1007/978-3-030-96709-3, 2022.

Fiddes, J. and Gruber, S.: TopoSUB: a tool for efficient large area numerical modelling in complex topography at sub-grid scales, Geoscientific Model Development, 5, 1245–1257, https://doi.org/10.5194/gmd-5-1245-2012, 2012.

Girotto, M., Musselman, K. N., and Essery, R. L. H.: Data Assimilation Improves Estimates of Climate-Sensitive Seasonal Snow, Current Climate Change Reports, 6, 81–94, https://doi.org/10.1007/s40641-020-00159-7, 2020.

Guidicelli, M., Aalstad, K., Treichler, D., and Salzmann, N.: A Combined Data Assimilation and Deep Learning Approach for Continuous Spatio-Temporal SWE Reconstruction from Sparse Ground Tracks, https://doi.org/10.2139/ssrn.4489553, 2023.

Magruder, L., Brunt, K., Neumann, T., Klotz, B., and Alonzo, M.: Passive Ground-Based Optical Techniques for Monitoring the On-Orbit ICESat-2 Altimeter Geolocation and Footprint Diameter, Earth and Space Science, 8, e2020EA001 414, https://doi.org/10.1029/2020EA001414, _eprint: https://onlinelibrary.wiley.com/doi/pdf/10.1029/2020EA001414, 2021.

Markus, T., Neumann, T., Martino, A., Abdalati, W., Brunt, K., Csatho, B., Farrell, S., Fricker, H., Gardner, A., Harding, D., Jasinski, M., Kwok, R., Magruder, L., Lubin, D., Luthcke, S., Morison, J., Nelson, R., Neuenschwander, A., Palm, S., Popescu, S., Shum, C. K., Schutz, B. E., Smith, B., Yang, Y., and Zwally, J.: The Ice, Cloud, and land Elevation Satellite-2 (ICESat-2): Science requirements, concept, and implementation, Remote Sensing of Environment, 190, 260–273, https://doi.org/https://doi.org/10.1016/j.rse.2016.12.029, 2017.

Mazzolini, M., Kristoffer, A., and Désirée, T.: Inputs (forcing, observations and config file) for the experiments included in "Spatio-temporal snow data assimilation with the ICESat-2 laser altimeter"., https://doi.org/10.5281/zenodo.11176897, 2024.

Mott, R., Vionnet, V., and Grünewald, T.: The Seasonal Snow Cover Dynamics: Review on Wind-Driven Coupling Processes, Frontiers in Earth Science, 6, URL https://www.frontiersin.org/articles/10.3389/feart.2018.00197, 2018.

Neuenschwander, A., Guenther, E., White, J. C., Duncanson, L., and Montesano, P.: Validation of ICESat-2 terrain and canopy heights in boreal forests, Remote Sensing of Environment, 251, 112 110, https://doi.org/10.1016/j.rse.2020.112110, 2020.

Neumann, T. A., Martino, A. J., Markus, T., Bae, S., Bock, M. R., Brenner, A. C., Brunt, K. M., Cavanaugh, J., Fernandes, S. T., Hancock, D. W., Harbeck, K., Lee, J., Kurtz, N. T., Luers, P. J., Luthcke, S. B., Magruder, L., Pennington, T. A., Ramos-Izquierdo, L., Rebold, T., Skoog, J., and Thomas, T. C.: The Ice, Cloud, and Land Elevation Satellite – 2 mission: A global geolocated photon product derived from the Advanced Topographic Laser Altimeter System, Remote Sensing of Environment, 233, 111 325, https://doi.org/10.1016/j.rse.2019.111325, 2019.

Revuelto, J., Azorin-Molina, C., Alonso-González, E., Sanmiguel-Vallelado, A., Navarro-Serrano, F., Rico, I., and López-Moreno, J. I.: Meteorological and snow distribution data in the Izas Experimental Catchment (Spanish Pyrenees) from 2011 to 2017, Earth System Science Data, 9, 993–1005, https://doi.org/10.5194/essd-9-993-2017, publisher: Copernicus GmbH, 2017.

Sturm, M., Goldstein, M. A., and Parr, C.: Water and life from snow: A trillion dollar science question, Water Resources Research, 53, 3534–3544, https://doi.org/10.1002/2017WR020840, _eprint: https://onlinelibrary.wiley.com/doi/pdf/10.1002/2017WR020840, 2017.

Treichler, D., Mazzolini, M., Piermattei, L., Webster, C., and Girod, L.: Spaceborne snow depth measurements from ICESat-2 laser altimetry, EARSel workshop 2023: Remote Sensing of the Cryosphere. Bern, Switzerland, URL https://www.earsel.org/SIG/Snow-Ice/files/Abstracts_EARSEL_2023_FINAL.pdf, 2023.

---

## Author Comment (AC2)

**General Comments:**

SMALL CAPS: COMMENT # 1.1

> *This study investigates using ICESat-2 satellite data to improve FSM2 simulations. The authors employ the DA proposed by Alonso-Gonzales et al. (2023) with a spatial propagation of the sparse data points from ICESat-2 that tries to compensate the fact that ICESat-2 data are acquired in profiles with many temporal and spatial gaps. They perform three experiments to assess the effectiveness of assimilating different data types snow cover area, from Sentinel-2, snow depth, from ICESat-2 or both. The reported findings indicate that by incorporating snow cover area data alongside snow depth from ICESat-2 led to the most accurate snowpack simulations.*

**Reply:**

We appreciate the Reviewer's interested approach to the paper and are grateful for the constructive suggestions that have helped us to improve the study. We kindly point out that the aim of the study is not primarily to improve the FSM2 simulations in the study catchment per se, but to present a newly developed method that allows the joint assimilation of snow cover data and novel ICESat-2 snow depth profiles that are sparse in time and space and thus currently of very limited use for snow modelling. The method presented here is designed to be globally applicable. ICESat-2's satellite-derived profiles have a very different nature and greater uncertainty than the experiments assimilating subsampled, within-catchment drone-based snow depth data presented by Alonso-González et al. (2023). Our method builds on this earlier work and the spatio-temporal data assimilation method therein to show how actual satellite-derived snow depth data from ICESat-2 can be propagated/transfered in space and time from outside the catchment using to yield spatio-temporally complete reconstructions of the full snowpack state in the catchment. Moreover, we also show (to the best of our knowledge) for the first time how to perform joint spatio-temporal assimilation of both ICESat-2 snow depth and Sentinel-2 fSCA so as to exploit the highly complementary nature of these two types of observations (Gascoin et al., 2024). In addition, we provide a thorough uncertainty quantification of the respective simulations, which is not trivial in the case of spatially propagated information from observations located near but outside the study area. The following provides a point-by-point response to the Reviewer's Comments.

SMALL CAPS: COMMENT # 1.2

*The authors leverage a data assimilation system (MuSA) from a previous work (Alonso-González et al., 2023). Moreover, the methodology for spatializing the ICESat-2 data (a point of significant interest) builds upon concepts presented in the previous work, but without substantial further development. While the initial findings are interesting, their persuasiveness could be strengthened through further analysis. This deeper exploration would allow the research to culminate in a more robust and impactful paper.*

**Reply:**

We agree that the core simulations of this paper relies on the MuSA assimilation system, which was developed in previous work (Alonso-González et al., 2022, 2023). However, we disagree with the rest of the Reviewer's statement while acknowledging that the novelty of this study could have been better emphasized in the paper. As such, we would like to highlight the differences and the method development we have accomplished to make these new experiments possible:

- Despite our short description in Section 3.1, a substantial effort was necessary to treat the ICESat-2 elevation observations in order to retrieve snow depth observations, as there is not an established method to do that. Careful data curation was required to eliminate outliers so that the observations could be ingested into a data assimilation system.

- To the best of our knowledge, this is the first work that goes beyond validating ICESat-2 data with snow depth measurements from another source, but instead assimilates them to constrain a snow model.

- The MuSA assimilation system was updated in order to spatially propagate information coming from a subset of the observations while doing a so-called joint data assimilation (assimilating multiple different types of observations). Because of the way we define a neighbourhood, a large number (many hundreds) of cells with fSCA-observed cells would be included in the neighbourhood. The computational burden would then increased drastically due to the resulting large input/output operations to access all neighboring fSCA predictions and observations for every Kalman update. The previous MuSA system, without the upgrade done in this study, would have required the resources of a larger supercomputer for such a joint assimilation. The updated assimilation method was implemented through modifications in the spatial_MuSA module of the system, changing the subset of observations that are considered to create a neighbourhood of pixels for domain localization that are considered for the state update. See the pull requests 14-17 in the MuSA repository. (https://github.com/ealonsogzl/MuSA/pull/14)

- For this work, we used the MuSA system to perform joint assimilation in a spatio-temporal setting. While the MuSA system has already been used to perform joint assimilation (Alonso-González et al., 2022), there the observations were land surface temperature and fractional snow-covered area but both were spatially complete products so this was done for a single pixel in a purely temporal ("1D", or "embarrassingly parallel") setting without any spatial propagation of information. Here we instead jointly assimilated snow depth and fSCA in a spatio-temporal setting whereby in the previous works snow depth was the only observation assimilated in experiments with spatial propagation. To the best of our knowledge, this is a completely novel snow data assimilation approach. In the described experiments, the two observation types (fSCA and snow depth) that we either jointly or individually assimilate have a different spatial coverage. This made calibrating the relative accuracy of the observations and the selection of the spatial propagation parameters such as the length scale a relatively tough exercise since these hyperparameters both have a considerable effect on the relative weight of the two sets of observations influencing the state of the simulations as well as on the spatial distribution of the simulation.

We realise that we may have undercommunicated the methodological development and MuSA updates required to achieve the results and framework presented in this study. In the revised manuscript, we better emphasize in the Introduction, in the Methods and in the Conclusions the novelty of the approach and how it differs from previous work.

COMMENT # 1.3

*While the current findings are interesting, further analysis could significantly enhance their persuasiveness. Consider incorporating other inputs data that only ERA-5 (also derived from in-situ) and also move to other (larger) catchment where distributed HS are available (e.g., ASO data if ICESat-2 data are available or Dischma in Switzerland) to solidify the results.*

**Reply:**

We are grateful for your acknowledgment of the interest of our findings and approach. To a limited extent we agree that further analysis would enhance their persuasiveness. We would like to address your suggestions on adding different forcing data and other study sites separately.

**Study site:** we appreciate your suggestion to experiment in larger catchments with distributed snow depth available. However, we propose to keep this as the sole study

site because of the acknowledged interest of the results shared by you and the other Reviewer, and expand the analysis on the utility of ICESat-2 observations to infer snow spatial-distribution point of view in the future (as suggested in Section 5). To our best knowledge, the locations you suggest are much larger and have not been surveyed by airborne instruments with the same temporal frequency available in the current study area (around 12 acquisitions in the studied season). The lower temporal resolution of distributed snow depth dataset could limit the solidity of the analysis on accumulation and ablation season. Moreover, high resolution and large scale DA exercises require substantial further model development or changes to the output resolution in order to be computationally affordable (e.g., not to simulate in a fully distributed manner or at a much coarser resolution). We therefore consider this possibility worthy of a complete new and separate study. Considerations about this are added in the Discussion.

**Forcing data:** the second suggestion is about incorporating in the analysis other input data than only ERA5, such as data derived from in-situ observations. We acknowledge that, if the scope would be to achieve the best possible snow simulation of this specific basin, it would certainly be the better option to use a continental or even national reanalysis together with in-situ data from a local meteorological station. However, we underline that this study was designed to showcase a globally applicable workflow where high-resolution forcing data is usually lacking – and the assimilation of observations is used to achieve distributed result maps. Note that one of the main motivation for using satellite DA in snow modeling is precisely to fix errors in the forcing data, so it is particularly worth demonstrating that these methods are able to work with coarse globally available meteorological forcing in line with many previous snow DA studies (e.g. Fiddes et al., 2019; Alonso-González et al., 2022, 2023). Moreover, several other DA studies are based solely on one global reanalysis as input forcing (in these cases MERRA or MERRA-2) such as Cortés et al. (2016) or Liu et al. (2021). Hence, we used the current state-of-the-art (in terms of resolution and accuracy) *global* atmospheric reanalysis ERA5. Its original spatial resolution clearly misses the hillslope scale heterogeneity, this we obtain partly through a preliminary topographic downscaling routine (TopoSCALE; Filhol et al., 2023) but mainly via the information contained in the observations that we assimilate. We considered this a sufficiently general prior knowledge of the seasonal snow evolution to be able to claim global applicability of this workflow, as it will arguably be most useful for other (less studied) areas where large knowledge gaps on snow amounts are existing and no regional forcing data (let alone in-situ observations) exist. In our opinion, adding another reanalysis to the experiments would confuse the reader and not add much value to the study for the following reasons:

- Starting from a higher quality prior knowledge of the seasonal snow evolution would leave the reader with the question: is this approach of any value for remote places (e.g. Central Asia) where such improved prior information is not available?

- The hillslope processes (100 m scales and smaller) we aim at simulating would not be represented in any higher resolution atmospheric reanalysis even using costly state-of-the art convection permitting atmospheric models which are at best at the kilometer rather than the hillslope scale.

We realise that we may have not emphasized enough that the suggested workflow was designed to be globally applicable. In the revised manuscript, we better underline this aspect when motivating the methodological choices both in the Introduction as well as in the Methods.

COMMENT # 1.4

*Revisit the scientific questions the paper aims to address. Sharpening these questions will guide the research and ensure the experiments directly address them.*

**Reply:**

Good point, we previously only stated a hypothesis at the end of Section 1. We now replace it with three specific research questions as we agree this can guide the reader to understand the idea behind our experiments.

**Changes:**

The two datasets have complementary features: ICESat-2 retrieves snow depth directly, but only along profiles; while fSCA has an indirect  relationship with snow depth, but  is spatially distributed.  The novel scientific questions we aim to answer are:

- Can information from sparse snow depth retrievals from ICESat-2 along profiles be used to provide information about average catchment-scale snow depth and its complete spatial distribution?

- Is assimilating sparse ICESat-2 snow depth retrievals better than more commonly used fSCA observations derived from optical satellites?

- Is ensemble-based DA able to able to leverage information from both observation types when jointly assimilating both fSCA and sparse snow depth observations?

**COMMENT # 1.5**

*Section 3.5 appears to hold the core of the paper contribution. Dedicating more space and development to this section would allow for a more thorough exploration and potentially lead to more impactful conclusions.*

**Reply:**

We fully agree with both the Reviewers' Comments stating that a deeper explanation of how information from observations is propagated in space is needed. The same suggestion came from Reviewer 1 in Comment 1.7. We propose to expand of Section 3.5 in the revised version of the manuscript to guide the reader better concerning the spatial propagation/transfer of information. See Comment 1.7 in this document for more details.

**Detailed comments:**

**COMMENT # 1.6**

*The introduction of the paper could benefit from being condensed and sharpened. Focus on presenting the key scientific questions, the research aims to answer, and clearly outlining the paper main novelty (difference with previous works). This will ensure the experiments directly target those questions and guide the research direction. The core innovation of the paper lies in applying the DA method (from Alonso-Gonzales et al. 2023) to ICESat-2 data. The unique approach for propagating spatial and climatological information holds significant promise. However, further development of this methodology is necessary, particularly regarding the justification for using data outside the area of interest for analyzing snow accumulation and redistribution (see next points).*

**Reply:**

Agreed, we have revised and shortened the introduction according to the structure proposed by the Reviewer as well as Comments from Reviewer 1, and also added research questions that the paper aims at answering. We take care to explain why we use snow depth observations outside the study area, which we find is one of the strengths of the proposed approach as it shows the utility of the sparse observations measured by the satellite ICESat-2 profiles. Our study catchment has only nearby ICESat-2 observation available, and the situation will be the same for many other

catchments of interest around the world, but the utility of ICESat-2 in snow modelling is still of interest.

<small>COMMENT # 1.7</small>

*Figure 3, potentially the paper core novelty, requires a more detailed explanation. From the scatterplot, it appears there might be a weak correlation between snow depth and CSMD (and TPI24). However, the relationship between snow depth and Sx200 seems less clear, potentially indicating no significant correlation. At least this is my understanding with the provided text. If this is not correct, I suggest a clearer description to enhance reader comprehension explicitly guide them through the correct explanation.*

**Reply:**

We originally included the Winstral index (Sx) because both Revuelto et al. (2014) and Mendoza et al. (2020), who carried out studies about the snow spatial distribution in the Izas experimental catchment, recommended the adoption of the Winstral index in addition to TPI when predicting snow depth. When exploring what dimensions to choose, we analysed their interplay in an explorative way and found that although Sx does not exhibit a linear correlation with SD, in combination with CSMD this dimension is a relevant predictor of SD (e.g. no low SD observations with a low Sx). However, due to the considerable cost of running the DA experiments we did not perform a complete factorial exploration of all possible feature dimensions. Nonetheless, following this particularly astute Reviewer Comment about the weak correlation between Sx and snow depth, we have now repeated the ensemble simulations for all DA experiment runs excluding Sx from the dimensions of the feature space. We were positively surprised to see a slight improvement in the results. We believe that in our case, the inclusion of that index is not beneficial because of the large amount of ICESat-2 observations that were located in an area with negative Winstral index, a characteristic shared with only few cells in the drone domain. It is possible to see this in the first submission's Figure 3. It seems that limited representativity of ICESat-2 data for the catchment topography in terms of Sx was leading to smaller correlation values and, consequently, a small influence of the observations for the cells of the simulated domain.

Hence, we propose a large change of Section 3.5 as you suggest in Comment 1.5, to guide the reader in understanding the spatial propagation of information. We removed Sx from the predictors and also add Figure 1 (see below) to the manuscript. The Figure exemplifies a situation where a cell in the experimental catchment with drone data – depicted in panel b) with a cross – has to be updated. The solid points in the scatterplot are selected to be part of the neighbourhood, and all of them have

[Figure]

Figure 1: *Panel a): scatterplot depicting the position of the cells from the drone maps in the feature space. This space – created with TPI and CSMD – is adopted to define the similarity between cells. The points are colored according to the snow depth observed with the drone. Panel b): ICESat-2 snow depth observations in the extended catchment, displayed in feature space, with snow depth-based coloring. The cross represents one cell from the drone domain where a snow depth of 150 cm was measured. The solid points are ICESat-2 data points included in the neighbourhood for this cell, with their size proportional to the correlation ρ.*

influence on the Kalman update (see step 11 in Algorithm 1 of Alonso-González et al., 2023), used to update the local ensemble of the target grid cell. As cells closer in feature space to the target cell should have a larger influence, their ρ is larger, which can be appreciated by looking at the size of the scatter points.

COMMENT # 1.8

*Please revise the text from L276-282 to make more clear (and less compressed).*

**Reply:**

This part of the text has been extended as part of the answer to Comment 1.5. See above the answer to Comment 1.7 for more details.

COMMENT # 1.9

*The paper relies solely on ERA-5 data for atmospheric forcing. While ERA-5 is a valuable product, acknowledging the existence of other models with potentially significant output variability (up to 100%) would strengthen the main message of the paper that ICESat-2 data can be useful and in which situation. A discussion on why ERA-5 was chosen over other op-*

*tions would be beneficial. However, for a more robust understanding of the proposed method I suggest incorporating data from at least a couple of additional models is recommended. This comparative analysis would highlight the method sensitivity to different forcing data. In particular, given the small target catchment area (and the high target resolution of 20m), exploring the use of spatially distributed data from nearby in-situ stations could be highly valuable. This would provide a more realistic scenario: not sure the first choice to simulate 5ha at a resolution of 20m in an experimental catchment in a European mountain range is starting from a 30km ERA-5 data.*

**Reply:**

We thank the Reviewer for the suggestion of a better argumentation for choosing ERA5 as the (only) source for forcing data. We point to the answer to Comment 1.3 and will not repeat the arguments here.

COMMENT # 1.10

*The results of experiment D are puzzling (at least to me in the present form). While Figure 3 (and related text) suggests that ICESat-2 data captures the relationship between SD distribution, topography, and climatology using this data alone in experiment D appears to yield inaccurate snow patterns, whereas it helps in experiment J when used together with Sentinel-2, what is the main mechanism behind this behavior?*

**Reply:**

We also find this result intriguing and did not find a simple answer. Below, we outline what factors likely contribute to the relatively poor result for experiment **(D)** but improved performance of experiment **(J)**. The spatial patterns we see in the distributed maps in the Results are governed by the features that design the spatially correlated prior. We agree that in experiment **(D)**, the snow patterns are inaccurate – but the basin average snow depth is greatly improved. The spatial patterns are a result of the features and their relative weights, which are the hyperparameters of the prior that were chosen from various tested combinations/variations, but not inferred or optimized. Adding fSCA observations – and hence moving to experiment **(J)** – adds to the local updates for the cells in the drone catchment a cumulative information about the accumulation and melt processes. Melt-out patterns are reproduced into snow patterns in the peak-SWE maps we show, while the snow depth profile provides information to adjust and improve absolute snow depth values.

COMMENT # 1.11

*A crucial evaluation metric for DA methods is computational time. The paper should explicitly report and analyze this metric, ideally providing a detailed profile for each operational step.*

**Reply:**

The simulations were run on a local server from the Department of Geosciences of the University of Oslo. It is equipped with a 1TB RAM and 40 processors were used for this task. However, there was high variance in computational time depending on the varying load on the server. For the three experiments the computational time was similar and at best it took 7 hours, or at worst three days. The computational cost depends on the GC parameter and on the density of observations, so it is hardly comparable to simulations in other sites. Moreover, we note that the current implemetation of MuSA is a wrapper around the Fortran implementation of FSM2. Simply improving FSM2, for example by translating it to the Python programming language, or other software-side improvements related to the observations use might greatly improve the computational time. Further development of MuSA with regard to computational efficiency is planned.

We acknowledge that computational time is a crucial factor for readers to know about the applicability of the method, and the order of magnitude will be mentioned in the revised manuscript in Section 3.6. However, technical differences in the implementation can make the computational time vary dramatically (see above). In this publication, the focus is on the scientific questions related to the utility of ICESat-2, and we prefer to avoid a lengthy technical sidetrack.

COMMENT # 1.12

*Fig 7 can you add the drone maps? Beside demonstrating a significant improvement in an accuracy score, the scientific community is starting to become interested in understanding how realistic snow distributions become when observations at high resolution are assimilated into models. The paper could benefit from a stronger emphasis on this aspect.*

**Reply:**

Two drone maps will be added to Fig 7, depicting the average snow depth for both the accumulation and melting seasons, so that the reader can compare the results with the absolute amounts.

COMMENT # 1.13

*While Figure 7 presents the CRPS for various scenarios, a deeper analysis could help isolate the contribution of ICESat-2 data. The similarity between the CRPS with ICESat-2 and the prior suggests limited influence of ICESat-2. However, the simultaneous improvement in J (potentially reflecting the contribution of ICESat-2) is intriguing and puzzling at the same time. Can you better comment on this?*

**Reply:**

We thank the Reviewer for the comment. For our experiments, one should note from the time series of Figure 4 that the validation maps average (black points) lie very close to the median of the prior (gray lines) during the accumulation period, hence the prior simulation already provides an accurate (albeit not precise) simulation in terms of average snow depth for this period. As a consequence, a gain on the CRPS score in the accumulation season is in our experiments harder to achieve than in the melting period, as there the prior average snow depth is quite far from the validation points and an improvement is thus easier to improve. In experiment **(D)**, there is a substantial improvement given by the diminishing of the spread of the ensembles. This is shown in the panel a) of the first submission's Figure 5 in terms of basin-average snow depth, which corresponds to a CRPS improvement of 14%. We agree that it's puzzling that this improvement is not better than what we achieve with fSCA-only assimilation. In the fSCA case, the basin average snow depth does not improve (see Figure 4, panel a)), where the blue trajectories overestimate the validation black points overall), but the reduction in CRPS is caused by a better relative spatial distribution of snow depth. The latter is visible from the similarity of panels b) and c) of Figure 4 as well as from the mostly uniform CRPS values in the spatially distributed map for experiment **(C)** in Figure 7. This is point is crucial: the improvements obtained by assimilating spatially complete fSCA and sparse snow depth are complementary, hence the large improvement we see in experiment **(J)**. We acknowledge it is important to underline this crucial point and we add this arguments in the discussions of the revised manuscript.

COMMENT # 1.14

*L430: the fact that SCA assimilation doesn't improve the simulation during the accumulation period could be attributed to the fact that the area of interest is having 100% snow cover?*

**Reply:**

The assimilation of fSCA using ensemble-based smoothers is a topic which has received many studies (e.g. Girotto et al., 2014; Margulis et al., 2015; Aalstad et al., 2018; Fiddes et al., 2019; Alonso-González et al., 2021). If the assimilation method acts as a filter, it acts sequentially: the observations modifies the state of the simulation at the current time, and the past (relative to the current observation) is not affected. Otherwise, for a batch smoother, (as in the presented experiments herein and references above), the information from an observation can also propagate backwards in time, as all the observations in the current water year are assimilated at once. It has previously been convincingly shown that fSCA assimilation can improve the seasonal snow simulations also in the accumulation season if such a smoother is employed because fSCA observations contain cumulative information about both accumulation and melting processes, indeed this is the key behind state-of-the-art probabilistic snow reanalyses (e.g. Margulis et al., 2016) and the earlier deterministic snow reconstruction techniques Girotto et al. (2014). However, since the information is integrated over both accumulation and melting-related parameters, equifinality problems can arise. That means there is not enough information in the fSCA observations to infer all the perturbation parameters, and very heterogeneous sets of parameter can be used to reconstruct the states as they're observed. This is the likely cause of the missing improvement in the fSCA experiment in terms of catchment-average snow depth, but note that there is an improvement in the CRPS score.

COMMENT # 1.15

*L439: this does not seem true to me (at least in the present form).*

**Reply:**

We agree, the sentence was a leftover from an earlier version of the results section. The paragraph will be changed:

**Changes:**

…As Figure 5 shows, this information leads to a more precise reconstruction of the catchment-average peak-SWE compared to experiment **(C)**. This demonstrates that the  spatial transfer of information method succesfully relates snow depth and the features, but only when averaging over the whole basin, as the

 relative spatial patterns of the simulation only partially match those of the validation maps (panels b) and c), Figure 5).

COMMENT # 1.16

*L442: this is in contradiction with L253 where it is stated that the snow depth is strongly governed by topography, which is also the main hypothesis why the proposed approach has been applied. This ping pong effects makes it challenging to understand the overall benefit of ICESat-2 data (and the validity of the presented results).*

**Reply:**

We agree that this is a contradiction, and we propose to modify both the sentences to remove the challenges for the reader. The consideration at line L442 will be modified to include that part of the feature space occupied by the drone domain is not covered by ICESat-2 observations.

**Changes:**

L253: … As the relative snow depth's spatial distribution  is strongly governed by topography, …

L442: …  The observations we use in this experiment are not direct measurements in the  experimental catchment, so this result is in the end not surprising . While the entire area experiences the same general snow conditions there are local differences which can be only partially captured with a low dimensional space, as TPI and CSMD do not fully characterize the snow depth distribution. For example, single cells with extreme values located in the  experimental catchment might not be similar in terms of  TPI and CSMD to only cells observed by ICESat-2 with as extreme snow depth values, but also to medium snow depth observations, not getting the ideal update.

COMMENT # 1.17

*L452: speculative, please revise it.*

**Reply:**

We rephrased.

**Changes:**

 As the ICESat-2 satellite will potentially collect data until the mid-2030s, and the snow science community is eager to keep on testing its potential to evaluate mountain water resources, this dataset has the potential become a functional tool for water managers to estimate the maximum seasonal snow accumulation. However, especially within an operational snow hydrological forecasting context (Mott et al., 2023), there is a clear need to reduce the processing time of the geolocated photon low level data which currently takes months.

COMMENT # 1.18

*L484: 20 m is not hyper resolution.*

**Reply:**

We acknowledge that the level of spatial resolution is a term relative to the context. For example, in climate modelling hyper-resolution is kilometric (Wood et al., 2011), while in non-ensemble forest snow modelling hyper-resolution is submetric. In the context of snow and hydrology ensemble-based DA where a cell is simulated numerous times, recent studies have defined their resolution as hyper for cell sizes well below 100 m (Fiddes et al., 2019; Alonso-González et al., 2023). In contrast, snow DA simulations with cell sizes of about 100 m typically define their spatial resolution as high (i.e., one level coarser than 'hyper') (Margulis et al., 2016; Girotto et al., 2020). Moreover, high resolution is also applied in recent literature to snow DA reconstructions at kilometric grid cells (Oaida et al., 2019; Brangers et al., 2023). At least within the context of snow DA, calling ensemble simulations at 20 m hyper resolution is warranted. In conclusion, we believe that this paper is not the location where an arguably ill-posed unified interdisciplinary definition of spatial resolution should be discussed. For such reason, we keep the term hyper in line with previous snow DA work.

COMMENT # 1.19

*References are generally ordered alphabetically.*

**Reply:**

The references were already ordered alphabetically by last name of the first author of each paper as is the norm, but we appreciate that the reference list is somewhat confusing given that it also included the full first names of all authors rather than just initials. The first names in the reference list will be replaced by initials in the final typeset version of the manuscript.

**REFERENCES**

Aalstad, K., Westermann, S., Schuler, T., Boike, J., and Bertino, L.: Ensemble-based assimilation of fractional snow-covered area satellite retrievals to estimate the snow distribution at Arctic sites, The Cryosphere, 12, 247–270, https://doi.org/10.5194/tc-12-247-2018, 2018.

Alonso-González, E., Gutmann, E., Aalstad, K., Fayad, A., Bouchet, M., and Gascoin, S.: Snowpack dynamics in the Lebanese mountains from quasi-dynamically downscaled ERA5 reanalysis updated by assimilating remotely sensed fractional snow-covered area, Hydrology and Earth System Sciences, 25, 4455–4471, https://doi.org/10.5194/hess-25-4455-2021, publisher: Copernicus GmbH, 2021.

Alonso-González, E., Aalstad, K., Baba, M. W., Revuelto, J., López-Moreno, J. I., Fiddes, J., Essery, R., and Gascoin, S.: The Multiple Snow Data Assimilation System (MuSA v1.0), Geoscientific Model Development, 15, 9127–9155, https://doi.org/10.5194/gmd-15-9127-2022, 2022.

Alonso-González, E., Aalstad, K., Pirk, N., Mazzolini, M., Treichler, D., Leclercq, P., Westermann, S., López-Moreno, J. I., and Gascoin, S.: Spatio-temporal information propagation using sparse observations in hyper-resolution ensemble-based snow data assimilation, Hydrology and Earth System Sciences, 27, 4637–4659, https://doi.org/10.5194/hess-27-4637-2023, 2023.

Brangers, I., Lievens, H., Getirana, A., and Lannoy, G. J. D.: Sentinel-1 snow depth assimilation to improve river discharge estimates in the western European Alps, preprint, Preprints, https://doi.org/10.22541/essoar.167690018.86153188/v1, 2023.

Cortés, G., Girotto, M., and Margulis, S.: Snow process estimation over the extratropical Andes using a data assimilation framework integrating MERRA data and Landsat imagery, Water Resources Research, 52, 2582–2600, https://doi.org/10.1002/2015WR018376, _eprint: https://onlinelibrary.wiley.com/doi/pdf/10.1002/2015WR018376, 2016.

Fiddes, J., Aalstad, K., and Westermann, S.: Hyper-resolution ensemble-based snow reanalysis in mountain regions using clustering, Hydrology and Earth System Sciences, 23, 4717–4736, https://doi.org/10.5194/hess-23-4717-2019, 2019.

Filhol, S., Fiddes, J., and Aalstad, K.: TopoPyScale: A Python Package for Hillslope Climate Downscaling, Journal of Open Source Software, 8, 5059, https://doi.org/10.21105/joss.05059, 2023.

Gascoin, S., Luojus, K., Nagler, T., Lievens, H., Masiokas, M., Jonas, T., Zheng, Z., and De Rosnay, P.: Remote sensing of mountain snow from space: status and recommendations, Frontiers in Earth Science, 12, https://doi.org/10.3389/feart.2024.1381323, publisher: Frontiers, 2024.

Girotto, M., Margulis, S. A., and Durand, M.: Probabilistic SWE reanalysis as a generalization of deterministic SWE reconstruction techniques, Hydrological Processes, 28, 3875–3895, https://doi.org/10.1002/hyp.9887, _eprint: https://onlinelibrary.wiley.com/doi/pdf/10.1002/hyp.9887, 2014.

Girotto, M., Musselman, K. N., and Essery, R. L. H.: Data Assimilation Improves Estimates of Climate-Sensitive Seasonal Snow, Current Climate Change Reports, 6, 81–94, https://doi.org/10.1007/s40641-020-00159-7, 2020.

Liu, Y., Fang, Y., and Margulis, S. A.: Spatiotemporal distribution of seasonal snow water equivalent in High Mountain Asia from an 18-year Landsat–MODIS era snow reanalysis dataset, The Cryosphere, 15, 5261–5280, https://doi.org/10.5194/tc-15-5261-2021, publisher: Copernicus GmbH, 2021.

Margulis, S. A., Girotto, M., Cortés, G., and Durand, M.: A Particle Batch Smoother Approach to Snow Water Equivalent Estimation, Journal of Hydrometeorology, 16, 1752–1772, https://doi.org/10.1175/JHM-D-14-0177.1, publisher: American Meteorological Society Section: Journal of Hydrometeorology, 2015.

Margulis, S. A., Cortés, G., Girotto, M., and Durand, M.: A Landsat-Era Sierra Nevada Snow Reanalysis (1985–2015), Journal of Hydrometeorology, 17, 1203–1221, https://doi.org/10.1175/JHM-D-15-0177.1, publisher: American Meteorological Society Section: Journal of Hydrometeorology, 2016.

Mendoza, P. A., Musselman, K. N., Revuelto, J., Deems, J. S., López-Moreno, J. I., and McPhee, J.: Interannual and Seasonal Variability of Snow Depth Scaling Behavior in a Subalpine Catchment, Water Resources Research, 56, e2020WR027343, https://doi.org/10.1029/2020WR027343, _eprint: https://onlinelibrary.wiley.com/doi/pdf/10.1029/2020WR027343, 2020.

Mott, R., Winstral, A., Cluzet, B., Helbig, N., Magnusson, J., Mazzotti, G., Quéno, L., Schirmer, M., Webster, C., and Jonas, T.: Operational snow-hydrological modeling for Switzerland, Frontiers in Earth Science, 11, URL https://www.frontiersin.org/articles/10.3389/feart.2023.1228158, 2023.

Oaida, C. M., Reager, J. T., Andreadis, K. M., David, C. H., Levoe, S. R., Painter, T. H., Bormann, K. J., Trangsrud, A. R., Girotto, M., and Famiglietti, J. S.: A High-Resolution Data Assimilation Framework for Snow Water Equivalent Estimation across the Western United States and Validation with the Airborne Snow Observatory, Journal of Hydrometeorology, 20, 357–378, https://doi.org/10.1175/JHM-D-18-0009.1, publisher: American Meteorological Society Section: Journal of Hydrometeorology, 2019.

Revuelto, J., López-Moreno, J. I., Azorin-Molina, C., and Vicente-Serrano, S. M.: Topographic control of snowpack distribution in a small catchment in the central Spanish Pyrenees: intra- and inter-annual persistence, preprint, Snow Hydrology, https://doi.org/10.5194/tcd-8-1937-2014, 2014.

Wood, E., Roundy, J., Troy, T., Beek, L., Bierkens, M., Blyth, E., Roo, A., Döll, P., Ek, M., Famiglietti, J., Gochis, D., and van de Giesen, N.: Hyperresolution Global Land Surface Modeling: Meeting a Grand Challenge for Monitoring Earth's Terrestrial Water, Water Resources Research - WATER RESOUR RES, 47, https://doi.org/10.1029/2010WR010090, 2011.

---

## Referee Report (RR1)

Review for "Spatio-temporal snow data assimilation with the ICESat-2 laser altimeter" by Mazzolini et al. under consideration in The Cryosphere

Summary: The authors clearly put a good amount of effort into addressing comments from the previous round of reviews and the manuscript is greatly improved. I do not have any major concerns about the manuscript but I have a number of minor comments that should be addressed by the authors prior to consideration for publication.

Minor Comments:
- Line 3: Remove ", despite being of great societal interest" because it makes the sentence somewhat awkward with where it is located in the sentence.
- Line 15: You currently end the abstract with stating that the skill score is improved by 19%. I generally recommend that abstracts include quantitative results if possible but this is hard to interpret without the reader knowing what the skill score means. You can keep this if you'd like but I recommend adding another sentence afterwards that summarizes the performance of the joint simulation so that the reader can easily understand the broader importance of the work. Are spatial variations in snow depth reproduced more accurately? Are temporal patterns in catchment-wide averages reproduced more accurately during the accumulation and/or ablation season? Providing this sort of information in the abstract will let the reader interpret the promise of the method even if they aren't sure how to interpret the skill score.
- lines 21-23: Instead of ending this sentence with a focus on measuring variability in snow "from space", I would say something like "at a watershed-scale using remote sensing methods" because you mention satellites, aircraft, and drones earlier in the sentence and the issue is really that we have problems getting accurate estimates across full watersheds.
- line 42-43: I found this sentence difficult to read. Consider revising this sentence and the one afterwards to be more straightforward with the fact that the data-assimilating intermediate snow model helps overcome issues with the use of coarse-resolution and potentially inaccurate large-scale atmospheric reanalyses for hydrologic forecasting.
- lines 47-61: Consider revising some of the sentences in this paragraph so that they do not focus as much on the authors of the referenced papers. When you start a sentence with the authors of a paper, you automatically focus on who did the work and not what the work tells you. For example, "Girotto et al. (2020) noted that most snow DA research – with a few exceptions – has focused on purely temporal DA…" could be revised to "Most snow DA research – with a few exceptions – has focused on…" without changing the message but taking the focus off of the citation.
- line 61: Remove the end of this sentence ",as recommended by comment 6 in Anonymous (2023) and by Gascoin et al. (2024)" because it really isn't needed.
- lines 82-87: The first sentence in this paragraph indicates that the paragraph is going to describe a hypothesis but that is not true. Please revise at least the first sentence.
- line 154: Typo "method"
- line 156: Typo "(Fiddes et al. 2019)"

- line 168: Change "a 3 m height" to "a height of 3 m"
- lines 173-178: For coregistration, you are working with an ICESat-2 track with snow across at least a portion of the domain. Did you only use photon from snow-free areas in your coregistration process? If so, how did you identify them as snow-free? If you used all the photons, that needs to be stated. Either way, it needs to be clear what photons were used because ultimately that impacts the accuracy of your snow depth estimates and therefore the model performance.
- Table 1: Typo "hyperparameters" in the caption. Also, I do not understand how the mean of the precipitation hyperparameter can be a negative number and outside the bounds that are provided. If that is not a typo, then these values should be explained more in the text because their interpretation is not straight-forward.
- line 435: It took me a few reads to understand this sentence. My interpretation is that fSCA in the accumulation season doesn't tell you much about snow depth because nearly the entire domain can be covered in snow and the depth can vary quite a bit but once you have melting snow and ground is exposed, the depths of the remaining snowpack are more consistent. If my interpretation is correct, try rephrasing so that this point is made more clearly.
- lines 464-466: After this sentence you should point out that ideally you would have incorporated another ICESat-2 profile based on the known improvement in simulations with ablation season observations, but you didn't have a good track from that time period.

---

## Author Response (AR2)

**REVIEWER #1**

COMMENT # 1.1

> *The authors clearly put a good amount of effort into addressing comments from the previous round of reviews and the manuscript is greatly improved. I do not have any major concerns about the manuscript but I have a number of minor comments that should be addressed by the authors prior to consideration for publication.*

**Reply:**

We appreciate the Reviewer's positive acknowledgment of our changes to the manuscript and thank them for their contribution and careful reviews. The suggested textual changes are helpful and greatly appreciated. The following provides a point-by-point response to the Reviewer's minor comments.

**Minor comments:**

COMMENT # 1.2

> *Line 3: Remove ", despite being of great societal interest" because it makes the sentence somewhat awkward with where it is located in the sentence.*

**Reply:**

We thank the reviewer for spotting this. We removed the sentence.

COMMENT # 1.3

> *Line 15: You currently end the abstract with stating that the skill score is improved by 19%. I generally recommend that abstracts include quantitative results if possible but this is hard to interpret without the reader knowing what the skill score means. You can keep this if you'd like but I recommend adding another sentence afterwards that summarizes the performance of the joint simulation so that the reader can easily understand the broader importance of the work. Are spatial variations in snow depth reproduced more accurately? Are temporal patterns in catchment-wide averages reproduced more accurately during the accumulation and/or ablation season? Providing this sort of information in the abstract will let the reader interpret the promise of the method even if they aren't sure how to interpret the skill score.*

**Reply:**

We appreciate the Reviewer's help in more carefully conveying the key messages in the abstract. We extended the abstract to detail better the joint assimilation results and the contribution from ICESat-2.

**Changes:**

Another encouraging finding is that adding the snow depth profiles to fractional snow-covered area observations leads to an accurate reconstruction of the snow depth spatial distribution. Evaluating the simulations with a set of independent drone-based snow depth maps using a probabilistic skill score, we find that for the accumulation season the joint assimilation's score improves by 19% the established approach of only assimilating fractional snow-covered area. The direct but incomplete snow depth information from ICESat-2 is a key constraint on simulated basin-average snow depth.

This study makes use of globally-available datasets and shows the promise of adopting ICESat-2's snow depth retrievals in seasonal snow modelling, especially when also assimilating complementary observations. In light of our encouraging results, more research with different experimental designs in varying snow conditions combined with continued methodological development is desirable to further catalyse the use of these retrievals in cryospheric and hydrologic applications.

COMMENT # 1.4

*lines 21-23: Instead of ending this sentence with a focus on measuring variability in snow "from space", I would say something like "at a watershed-scale using remote sensing methods" because you mention satellites, aircraft, and drones earlier in the sentence and the issue is really that we have problems getting accurate estimates across full watersheds.*

**Reply:**

We recognise the confusion in the sentence, and modify accordingly:

**Changes:**

Despite the many approaches involving satellite, airborne and drone sensors of various different types currently being used, accurately measuring the temporal and spatial variability in snow amount (i.e., mass or depth)  at watershed-scale with remote sensing is still a major scientific challenge (Gascoin et al., 2024).

COMMENT # 1.5

*line 42-43: I found this sentence difficult to read. Consider revising this sentence and the one afterwards to be more straightforward with the fact that the data-assimilating intermediate snow model helps overcome issues with the use of coarse-resolution and potentially inaccurate large-scale atmospheric reanalyses for hydrologic forecasting.*

**Reply:**

We simplified the structure of the thought as follows:

**Changes:**

 The main limitation for large-scale snow hydrology modelling is the accuracy of atmospheric forcing data (Raleigh et al., 2015). This applies especially  to spatially-distributed  setups where the forcing needs to be extracted from large scale atmospheric model outputs such as coarse-resolution (30 km) global atmospheric reanalyses (e.g. ERA5; Hersbach et al., 2020).

SMALL CAPS: COMMENT # 1.6

*lines 47-61: Consider revising some of the sentences in this paragraph so that they do not focus as much on the authors of the referenced papers. When you start a sentence with the authors of a paper, you automatically focus on who did the work and not what the work tells you. For example, "Girotto et al. (2020) noted that most snow DA research – with a few exceptions – has focused on purely temporal DA…" could be revised to "Most snow DA research – with a few exceptions – has focused on…" without changing the message but taking the focus off of the citation.*

**Reply:**

Thanks for this useful advice. We agree that the attention is on the authors of that paper, but in this case, we chose to do that as the cited paper is a review paper (Girotto et al., 2020) where an important limitation of current snow DA is highlighted. Changing the order would put this review (which we want to keep in the main sentence) on the same level as the few exceptions. We hence keep this particular phrase as is. However, we have changed another sentence in the paragraph which fits under the point made by the reviewer:

**Changes:**

...  A greater adoption of spatio-temporal multivariate DA is recommended (De Lannoy et al., 2022). ...

COMMENT # 1.7

*line 61: Remove the end of this sentence ",as recommended by comment 6 in Anonymous (2023) and by Gascoin et al. (2024)" because it really isn't needed*

**Reply:**

We acknowledge that the phrase could be fine without the ending as the Reviewer suggests, but we would like to keep this to show that the community has previously expressed the need for such experiments as those that we are carrying out in this work.

COMMENT # 1.8

*lines 82-87: The first sentence in this paragraph indicates that the paragraph is going to describe a hypothesis but that is not true. Please revise at least the first sentence.*

**Reply:**

Right, we change to:

**Changes:**

 An important assumption of this work, shared with other high-resolution DA assimilation studies, can be outlined as follows...

COMMENT # 1.9

*line 154: Typo "method"*

**Reply:**

Thanks for spotting, corrected.

COMMENT # 1.10

*line 156: Typo "(Fiddes et al. 2019)"*

**Reply:**

Corrected.

COMMENT # 1.11

*line 168: Change "a 3 m height" to "a height of 3 m"*

**Reply:**

We changed the text to make it more readable.

COMMENT # 1.12

*lines 173-178: For coregistration, you are working with an ICESat-2 track with snow across at least a portion of the domain. Did you only use photon from snow-free areas in your coregistration process? If so, how did you identify them as snow-free? If you used all the photons, that needs to be stated. Either way, it needs to be clear what photons were used because ultimately that impacts the accuracy of your snow depth estimates and therefore the model performance*

**Reply:**

We thank the reviewer for helping make this part of the methods clearer. The co-registration process does not use only the snow-free area, as there are not enough of these areas in the used DEM, but we make use of all the "selected" photons (according to the method defined in the lines 165-173). We modified as follows to explain that each beam is coregistered with the snow-off DEM.

**Changes:**

Before comparing the ATL03 photon events to the snow-free reference surface elevation it is necessary to co-register this dataset with the snow-off DEM.   We employ the Nuth-Kääb algorithm to obtain the horizontal shifts using as input all the selected ICESat-2 photon events and the snow-off DEM elevations (Nuth and Kääb, 2011), implemented in the xdem python library (Dehecq et al., 2021). Every beam is independently co-registered to account for different horizontal displacement. Individual vertical offsets (expected due to snow cover) are not removed; instead, we correct with a common offset

between the DEM all snow-free ICESat-2 data following established practice (Enderlin et al., 2022; Besso et al., 2024).

*Table 1: Typo "hyperparameters" in the caption. Also, I do not understand how the mean of the precipitation hyperparameter can be a negative number and outside the bounds that are provided. If that is not a typo, then these values should be explained more in the text because their interpretation is not straight-forward.*

**Reply:**

Thanks for the comment, we corrected the typo. The effect of a negative mean parameter of the associated Gaussian distribution for the logit-normal prior on the precipitation is to have a right-skewed (left leaning) density distribution for the precipitation perturbation. It is important to note that this is not the mean of the logit-normal distribution itself, but rather the mean of the associated (transformed) normal distribution. The logit-normal is usually specified this way since given its analytical form it makes more sense to define it in terms of the parameters of the associated normal distribution. In fact, there is no analytical expression for the mean or variance of the logit-normal distribution. This is described in detail in (e.g.) (Aitchison and Shen, 1980). It is nonetheless possible to analytically compute the median of the logit-normal by taking the scaled expit (inverse logit) transform of $\mu$ so we report this in the text. To better, albeit briefly, explain these details, we modified the paragraph where the prior perturbation extraction is explained.

**Changes:**

The perturbation parameters are time-invariant throughout the water year, and are extracted from a logit-normal distribution whose prior hyperparameters  can be seen in Table 1.  The prior perturbations are obtained by extracting samples from an associated normal distribution with mean $\mu$ and standard deviation $\sigma$. Subsequently, the inverse logit transform (sometimes called expit) with scaling is applied, resulting in a logit-normally distributed sample ranging from the lower to the upper bound (see section 3.3.1 in Aalstad et al., 2018). We choose the logit-normal distribution over a log-normal or a  normal distribution as the logit-normal restricts the perturbation within defined upper and lower bounds (shown in Table 1), in contrast to other distributions which would have respectively only one or no bounds (Aitchison and Shen, 1980). The nature of the perturbation is multiplicative for the precipitation (in part to prevent non-physical negative values) and additive for the other variables. The negative-mean parameter of the

associated normal distribution in the multiplicative perturbation parameter for the precipitation results in a right-skewed (left leaning) prior distribution with a median of 1.5.

**COMMENT # 1.14**

*line 435: It took me a few reads to understand this sentence. My interpretation is that fSCA in the accumulation season doesn't tell you much about snow depth because nearly the entire domain can be covered in snow and the depth can vary quite a bit but once you have melting snow and ground is exposed, the depths of the remaining snowpack are more consistent. If my interpretation is correct, try rephrasing so that this point is made more clearly.*

**Reply:**

We thank the reviewer for helping clarifying this. We inverted the order of the sentence and added a motivation to explain this.

**Changes:**

In experiment **(C)**, we simulate the snowpack and assimilated fSCA to create a baseline.  The fSCA generally shows lower correlation with snow depth earlier in the snow season (when the ground is fully covered by snow) compared to the  melting season (Girotto et al., 2020), but previous works show that assimilating fSCA can allow for an accurate reconstruction of peak SWE (e.g. Girotto et al., 2014). Indeed, experiment **(C)** shows high accuracy and precision especially towards the end of the season. As the experiments adopt a  smoothing approach, the melt-out pattern information contained in the fSCA observations is also propagated backward in time, and the posterior simulation offers a relatively accurate reconstruction for the peak-SWE: the validation is clearly close to the median ensemble spread, but the reconstruction for this part of the season is less precise compared to experiment **(D)**, and both less precise and accurate compared to experiment **(J)**.

**COMMENT # 1.15**

*lines 464-466: After this sentence you should point out that ideally you would have incorpo-*

*rated another ICESat-2 profile based on the known improvement in simulations with ablation season observations, but you didn't have a good track from that time period*

**Reply:**

We thank the reviewer for suggesting to add this consideration in the discussion.

**Changes:**

[revised manuscript text omitted]

**Reviewer #2**

**Comment # 1.1**

*The authors have clearly put considerable effort into revising the manuscript based on the reviewers' feedback. The revisions have resulted in a more focused presentation, and the clarification of several key steps has significantly improved the manuscript clarity. While the assimilation of ICESat-2 data represents a significant strength of this study, I believe further in-depth investigation could optimize its impact. Specifically, exploring the sensitivity of the assimilation results to factors such as the number and temporal distribution of ICESat-2 acquisitions, as well as the specific snow conditions (e.g., accumulation vs melting, or shallow vs. deep snowpack) at the time of acquisition, could yield even more robust results. Dedicating more research time to systematically analyzing these aspects might allow for a deeper exploitation of the unique capabilities of the ICESat-2 dataset for snow assimilation, potentially revealing optimal acquisition strategies for different scenarios. Furthermore, while the authors present a promising methodological approach, the current validation, limited to a single, small catchment in the Pyrenees, raises concerns regarding the generalizability and global applicability of the findings. Claiming global applicability would necessitate a significantly more geographically diverse set of experiments, encompassing a wider range of snow regimes, terrain complexity, and climate conditions (at least in my view). While the current results provide a valuable initial foundation, the study is still in the early stages of demonstrating broad applicability. Given that the exploration of ICESat-2 data assimilation in snow models is still in its nascent stages, clearly outlining these limitations and future research directions will be crucial for the manuscript impact and for guiding future research in the field.*

**Reply:**

We thank the Reviewer for contributing to the improvement of the manuscript through the previous review, and for acknowledging the effort the authors have put into improving the clarity and focus of the manuscript.

We agree that further in-depth investigation and methodology development is desirable in the future. In particular, wider experiments with different dates of acquisition, snow conditions will shed light on when ICESat-2 observations are the most valuable. Moreover, we agree that further methodological development is needed to use ICESat-2 observations on large-scale snow reanalysis.

We are grateful that the reviewer evaluates our methodological approach as promising. We see this as the first step towards a wider use of ICESat-2 for cryospheric applications, and think it would be presumptuous from our manuscript (the first using ICESat-2 observations in a DA setting) to give a final answer or to define one method that works for all the possible applications. We also want to push back a bit by saying

that this first step we took could have been much smaller. For example, the experiments could have been carried out at the point scale or along a single ICESat-2 profile. Instead, they were conducted in a fully spatio-temporal joint data assimilation setup at very high resolution for an entire catchment (not directly observed by ICESat-2). In the manuscript, we underline that the datasets used are available globally, and this was a deliberate choice because it was developed so that it could be applied in different regions where larger uncertainties in snow accumulations are present. We do not claim that this method is the way to improve snow modelling in all the world's regions, that would require further experiments. We do, however, both claim and demonstrate that assimilating ICESat-2 snow depth holds considerable promise and is worthy of continued research.

Following the Reviewer's suggestion, we added some considerations on the limitations of this work in the discussion.

**Changes:**

(line 476, in section 5.1)

As a recommendation, we add that it would be ideal to add more ICESat-2 observations later in the water year, to force the melting season better if that is available. Moreover, experimenting with different temporal and spatial distribution of the assimilated observations as well as with conditions of shallow or deep snow could clarify even more the utility of ICESat-2 in the context of seasonal snow modelling.
... (line 513, in section 5.2)

Moreover, the proposed method relies on a spatially correlated prior to propagate the sparse observations. This step requires the inversion of a squared matrix as large as the number of the simulated cells. Being this is a computationally expensive process, further research is recommended as approximations might be needed to extend the proposed methods to large basins. In addition, larger domains could be split to diminish the size of the matrix to invert and make this process feasible..

Moreover, we added that further experiments and methodological developement is needed in a last paragraph in the abstract.

**Changes:**

This study makes use of globally-available datasets and shows the promise of adopting ICESat-2's snow depth retrievals in seasonal snow modelling, especially when also assimilating complementary observations. In light of our encouraging results, more research with different experimental designs in varying snow conditions combined with continued methodological development is desirable to further catalyse the use of these retrievals in cryospheric and hydrologic applications.